# MuRating: A High Quality Data Selecting Approach to Multilingual Large Language Model Pretraining

**Zhixun Chen**[1][*]   **Ping Guo**[2]   **Wenhan Han**[3]   **Yifan Zhang**[2]   **Binbin Liu**[2]
**Haobin Lin**[2]   **Fengze Liu**[2]   **Yan Zhao**[2]   **Bingni Zhang**[2]   **Taifeng Wang**[2]
**Yin Zheng**[2,†]   **Trevor Cohn**[4]   **Meng Fang**[5,3,†]

[1]Hong Kong University of Science and Technology (Guangzhou) [2]ByteDance
[3]Eindhoven University of Technology [4]University of Melbourne [5]University of Liverpool

## Abstract

Data quality is a critical driver of large language model performance, yet existing model-based selection methods focus almost exclusively on English, neglecting other languages that are essential in the training mix for multilingual LLMs. We introduce MuRating, a scalable framework that transfers high-quality English data-quality signals into a multilingual autorater, capable of handling 17 languages. MuRating aggregates multiple English autoraters via pairwise comparisons to learn unified document quality scores, then projects these judgments through translation to train a multilingual evaluator on monolingual, cross-lingual, and parallel text pairs. Applied to web data, MuRating selects balanced subsets of English and multilingual content to pretrain LLaMA-architecture models of 1.2B and 7B parameters. Compared to strong baselines, including QuRater, FineWeb2-HQ, AskLLM, DCLM, our approach increases average accuracy on both English benchmarks and multilingual evaluations. Extensive analyses further validate that pairwise training provides greater stability and robustness than pointwise scoring, underscoring the effectiveness of MuRating as a general multilingual data-selection framework.

## 1 Introduction

Large Language Models (LLMs) have achieved remarkable performance across a wide range of tasks, and recent studies have consistently emphasized the critical role of high-quality pretraining data in driving these advances [5, 51, 7]. To improve data quality, various strategies have been adopted, such as deduplication [31, 1], heuristic and rule-based filtering [51, 45], and domain-aware sampling [65, 54]. While effective, these methods often rely heavily on manual heuristics and domain expertise, lacking a unified or principled framework for evaluating and selecting pretraining data. Moreover, they are typically applied as pre-defined or post-hoc filters, limiting their adaptability to downstream performance. In response, model-based data selection approaches have emerged, aiming to learn data quality judgments from examples or auxiliary supervision. These methods utilize different model architectures and data selection criteria. For instance, DCLM [33] trains a FastText classifier [27] using high-quality samples from OH2.5 and Reddit ELI5 as positive supervision, while treating Common Crawl web data as negatives. Other approaches such as AskLLM [52], QuRater [64], and the FineWeb-Edu [38] classifier employ prompt-based evaluation criteria using various LLMs to assess the quality of input samples.

---

[*]Work done during Zhixun's internship at ByteDance. Yin Zheng is the tech lead of multilingual LLM pretrain project at ByteDance.

[†] Corresponding Authors. Emails: yzheng3xg@gmail.com and Meng.Fang@liverpool.ac.uk.

Data selection beyond English remains an important challenge [13, 30]. While model-based data selection methods have demonstrated effectiveness in improving training quality, they have been developed almost entirely for English and are not explicitly designed or validated for non-English languages, leaving a critical gap in multilingual data quality assessment. As LLMs are increasingly applied in diverse linguistic contexts, there is a growing need for selection strategies that extend beyond English. A recent attempt Fineweb2-HQ [39] train language-specific raters using benchmark datasets as positive supervision and general pretraining corpora as negatives, following a strategy similar to DCLM [33]. However, this approach uses benchmark-derived data, posing a risk of test set contamination.

In this work, we introduce MuRating, a two-stage, translation-and-pairwise framework for multilingual data-quality estimation. MuRating begins by aggregating multiple state-of-the-art English raters via majority-vote pairwise comparisons, fitting a Bradley–Terry model [4] to learn a single, unified quality scorer. Next, it translates scored English document pairs into each of 17 target languages and construct monolingual, cross-lingual, and parallel pairs—projecting original preference labels onto translated comparisons and assigning neutral labels to parallel translations. Here, parallel pairs consist of identical content translated into two different languages, while cross-lingual pairs involve distinct texts written in different languages. This design yields one multilingual evaluator that preserves English-derived quality signals while remaining language-agnostic.

We apply MuRating framework to fine-tune a MuRater model to annotate English and multilingual web documents and select top 10% data to pretrain LLaMA-architecture [17] models of 1.2B and 7B parameters for validation. Compared to strong baselines—uniform sampling with 50% more data, QuRater [64], AskLLM [52], FineWeb2-HQ [39]—our selection yields an average gain of 1 to 3.4 points on twelve English benchmarks and 1.8 points on a diverse multilingual suite. We further assess translation fidelity via human evaluation, examine the impact of cross-lingual and parallel data, and compare different score transfer approaches.

Our contributions are as follows:[2]

- Unified English rater aggregation. We consolidate four distinct English quality raters via a Bradley–Terry pairwise framework, producing a single, robust scoring model.

- Translation-based multilingual transfer. We show how to project English pairwise judgments into monolingual, cross-lingual, and parallel pairs across 17 languages, enabling language-agnostic quality evaluation.

- Scalable pretraining gains. The results from both 1.2B and 7B model experiments demonstrate significant gains over state-of-the-art baselines across English and multilingual LLM benchmarks.

## 2 Related Work

**Data Selection.** Data selection is essential in constructing high-quality pretraining corpora for LLMs and typically falls into three main categories: deduplication, heuristic-based filtering, and LLM-guided quality evaluation. Early-stage deduplication removes exact or near-duplicate documents to minimize redundancy and enhance model generalization [31]. More advanced fuzzy and semantic methods filter syntactically or semantically similar content [25, 1], which is crucial at scale to avoid training instability and performance degradation [68, 51].

Heuristic filtering uses rules or lightweight models to exclude low-quality text, such as short, repetitive, or toxic content [30, 62, 47, 55]. While handcrafted heuristics can be effective, they often have limited generalization and inefficiency, prompting the use of simple classifiers, perplexity scores, or importance sampling [7, 59, 66, 42, 36, 35]. However, these approaches may unintentionally favor simplistic or repetitive content, which can diminish the diversity and informativeness of the dataset.

In contrast, LLM-guided quality scoring directly leverages language models to evaluate data along dimensions like factuality and coherence [18, 52, 34]. Frameworks such as QuRating [64] and FineWeb-Edu [38] prioritize educational content using multi-criteria assessment , while Dataman [48] and FIRE [67] extend this to domain-aware or reliability-sensitive filtering. Despite their

---

[2]Our code is available at: `https://github.com/aialt/MuRater`.

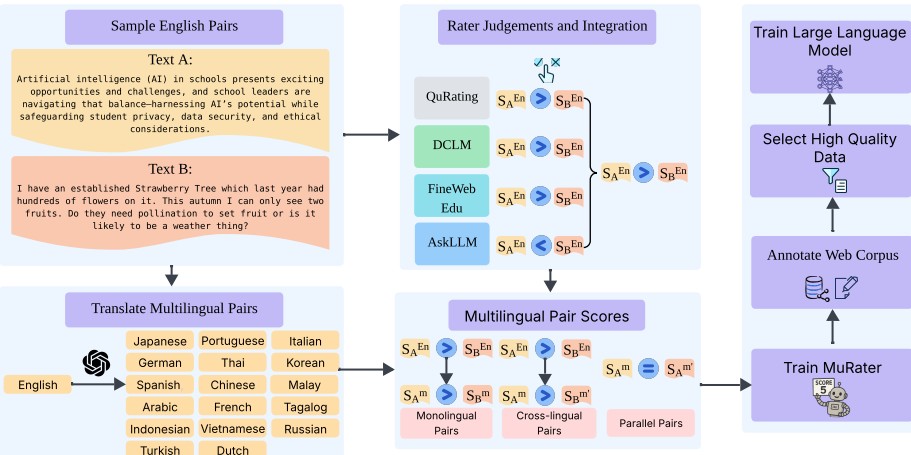

Figure 1: Overview of the MuRating pipeline: English document pairs are first annotated using various data selection methods and unified, then translated into multiple languages to create diverse multilingual pairs. These are used to train the MuRater model, which scores large-scale web data. The top 10% of scored data is selected to train an LLM, yielding superior performance compared to state-of-the-art sampling baselines.

advancement, recent approaches depend heavily on GPT-style judgments, potentially introducing model-specific biases.

**Multilingual Pretraining.** Efforts to construct multilingual datasets for multilingual LLM pretraining have followed similar strategies to those used for English, incorporating deduplication and heuristic-based filtering techniques. Prominent corpora such as mC4 [69], RedPajama [62], CulturalX [43], HPTL [12], and FineWeb-2 [46] leverage these methods to scale multilingual resources, ranging from a few dozen to thousands of languages, significantly enhancing cross-lingual performance in multilingual LLM pretraining. To mitigate the issue of data scarcity for low-resource languages, TransWeb-Edu [61] addresses this by translating high-quality English data into multiple languages. In addition, EMMA-X [20] introduces an EM-inspired framework that jointly learns cross-lingual semantic alignment and sentence representations from large-scale non-parallel multilingual data, while CLIMB [19] enhances multilingual capability by dynamically adjusting the data-mix ratio across languages. However, research on model-based data selection for multilingual LLM pretraining remains limited. Recently, an initial approach [39] introduced a model-based selection method to refine the FineWeb-2 dataset by training language-specific classifiers, using multilingual benchmark datasets as positive examples and web corpus data as negative examples. However, this method relies on the availability of high-quality samples from existing multilingual benchmarks, which may risk contaminating downstream evaluation tasks with biased data.

## 3 Methodology

Our approach consists of two stages: (1) consolidating multiple English-language quality raters into a single, unified scorer via pairwise comparisons, and (2) transferring the scorer to a multilingual setting through translation-based alignment and cross-lingual regularization. We introduce a two-step method: first, we integrate existing English corpus quality raters; second, we transfer their rating capability to a multilingual setting.

### 3.1 Integration of English AutoRaters

To consolidate quality judgments from multiple pre-existing English raters, we employ a pairwise comparison framework grounded in statistical preference modeling. Let $(t_A, t_B)$ denotes a pair of texts randomly sampled from a large corpus, and let $N$ be the set of raters. Each rater $n \in N$ assigns a scalar score to both texts, denoted as $S_A^n$ and $S_B^n$, reflecting the rater's estimation of the quality of $t_A$ and $t_B$, respectively.

We define a binary preference for each rater: if $S_A^n > S_B^n$, we consider that rater $n$ prefers text $t_A$ over $t_B$, and vice versa. If the two scores are nearly identical (i.e., $|S_A^n - S_B^n| < \epsilon$), we treat the preference as ambiguous and discard the pair from the training dataset. Based on the remaining valid preferences from all raters, we compute an empirical confidence score $P_{A>B}$ indicating how likely $t_A$ is preferred over $t_B$:

$$P_{A>B} = \frac{1}{|N|} \sum_{n \in N} \mathbb{I}[S_A^n > S_B^n], \quad P_{A>B} \in [0,1], \quad \text{where } |S_A^n - S_B^n| \geq \epsilon. \tag{1}$$

where $\mathbb{I}[\cdot]$ is the indicator function that equals 1 when the condition is true and 0 otherwise. if This score quantifies the relative preference strength of $t_A$ over $t_B$ across all raters.

To construct a large-scale preference dataset, we apply this scoring procedure across a wide set of sampled text pairs. This process yields a judgment dataset: $\mathcal{J} = \{(t_A, t_B, P_{A>B})\}$ consisting of text pairs and the estimated probability of preference.

To convert these pairwise comparisons into continuous scalar quality scores, we employ a learning framework based on the Bradley-Terry model [4]. Let $s_\theta(t)$ denote the learnable scalar quality score of text $t$, parameterized by $\theta$. We adopt a binary cross-entropy loss function, following the formulation proposed in [64], which is analogous to the reward model training paradigm in Reinforcement Learning from Human Feedback (RLHF) [44], but without incorporating user prompts or conditioning on input queries:

$$\mathcal{L}_\theta = \mathbb{E}_{(t_A, t_B, p_{B \succ A}) \in \mathcal{J}} \left[ -p_{B \succ A} \log \sigma(s_\theta(t_B) - s_\theta(t_A)) - (1 - p_{B \succ A}) \log \sigma(s_\theta(t_A) - s_\theta(t_B)) \right], \tag{2}$$

where $\sigma(\cdot)$ denotes the sigmoid function, and $p_{B \succ A} = 1 - P_{A>B}$ is the empirical probability that $t_B$ is preferred over $t_A$. This formulation encourages the model to assign higher scores to texts that are consistently preferred in the pairwise judgments.

After training, the model outputs a single scalar score representing the quality of each document. These scores are treated as logits over the dataset and are used for quality-based sampling, where a subset of high-quality texts is selected based on their relative scores.

## 3.2 Multilingual Data Quality Rater

### 3.2.1 Translation-Based Alignment of Multilingual Preferences

To extend data quality scoring from English to a set of target languages $M$, we adopt a translation-based strategy. Building on the scored English text pairs introduced in the previous section, we translate each document pair $(t_A^{en}, t_B^{en})$ into a target language $m \in M$. For each pair, we compute a confidence score $P_{A^{en} > B^{en}}$ following Equation 1, and then directly transfer this preference to the translated pair by assuming $P_{A^m > B^m} \approx P_{A^{en} > B^{en}}$.

We also experimented with the reverse approach—translating multilingual pairs into English, scoring them in English, and then projecting the scores back to the corresponding multilingual pairs. A comparison between the two setups is presented in the experiment section 4.2.1.

This assumption is based on the premise that translation preserves both the semantic content and the relative quality between text pairs. Prior work QuRating [64] highlights that pairwise comparisons offer increased stability when evaluating text quality. In multilingual settings, pointwise scoring—where absolute quality scores are assigned to individual texts—is more susceptible to subtle changes in tone or phrasing introduced during translation, which can compromise the consistency of the supervision signal. In contrast, pairwise supervision is inherently more robust to such translation-induced variations. As long as the relative ranking between the texts remains consistent (i.e., $t_A^{en}$ continues to be preferred over $t_B^{en}$ after translation), the corresponding translated pair $(t_A^m, t_B^m)$ remains a valid training example. This robustness makes pairwise comparisons a more reliable and effective framework for training quality evaluation models in multilingual contexts.

## 3.3 Cross-Lingual and Language-Agnostic Alignment

While the previous section addressed only in-language supervision—i.e., training on text pairs $(t_A^m, t_B^m)$ where both documents are in the same language $m$—this setup alone is insufficient to

guarantee language-agnostic scoring behavior. To promote consistency in quality assessments across languages, we augment our training dataset with both cross-lingual and parallel text pairs.

For cross-lingual pair construction, we generate mixed-language pairs by randomly translating $t_A$ and $t_B$ into different target languages, resulting in pairs of the form $(t_A^m, t_B^{m'})$ with $m \neq m'$. The original English pairwise preference score is then transferred to these cross-lingual pairs by assuming $P_{A^m > B^{m'}} \approx P_{A^{en} > B^{en}}$.

For parallel pair construction, we build semantically equivalent pairs to explicitly regularize the model's behavior. Given a text $t_A^m$ and its direct translation $t_A^{m'}$ into another language $m'$, we form the pair $(t_A^m, t_A^{m'})$ and assign a neutral preference score, i.e., $P_{A^m > A^{m'}} \approx 0.5$. This reflects the expectation that both texts, despite being in different languages, convey identical semantic meaning and should be treated as equally quality.

Formally, these neutral-pair constraints act as a regularization signal that aligns the model's internal representation of quality across languages:

$$\mathcal{L}_{\text{parallel}} = \mathbb{E}_{(t_A^m, t_A^{m'}) \in \mathcal{J}'} \left[ -\log \sigma \Big( s_\theta(t_A^m) - s_\theta(t_A^{m'}) \Big) - \log \sigma \Big( s_\theta(t_A^{m'}) - s_\theta(t_A^m) \Big) \right], \quad (3)$$

where $\mathcal{J}'$ is the datasets of parallel pairs. This formulation encourages the model to minimize score divergence between translations while still preserving the ability to differentiate documents of genuinely different quality in the broader training set.

### 3.3.1 Multilingual Rater Objective

The final loss function is a combination of the original pairwise loss from same-language and cross-language comparisons, along with the parallel text regularization term:

$$\mathcal{L}_{\text{total}} = \mathcal{L}_{\text{pairwise}} + \lambda \cdot \mathcal{L}_{\text{parallel}}, \quad (4)$$

where $\mathcal{L}_{\text{pairwise}}$ means the loss calcuated by equation 2 for both monolingual and cross-lingual pairs, $\lambda$ is a tunable hyperparameter balancing cross-lingual consistency and discrimination. This joint training approach allows us to construct a multilingual quality rater that is robust, consistent across languages, and sensitive to relative quality differences.

### 3.3.2 Training the Rater Model

To construct a high-quality multilingual rater, we begin with 300,000 English text pairs annotated using four rating methods. For GPT-4o-based annotation, we prompt the model in both directions—$(t_A, t_B)$ and $(t_B, t_A)$—multiple times to mitigate order bias, and compute the final confidence score $P_{A>B}$ by averaging the predicted preference probabilities. For other raters (AskLLM [52], FineWeb-Edu-Classifier [38], DCLM [33]), we collect their individual scores and derive pairwise preferences following Equation 1.

We then extend these English pairs into multilingual settings using GPT-4o for translation: 150,000 monolingual pairs, 150,000 cross-lingual pairs, and 75,000 parallel pairs $(t_A^m, t_A^{m'})$, with language proportions balanced across all target languages. The combined dataset—comprising English, monolingual, cross-lingual, and parallel examples—forms the final MuRater training set. We adopt QuRater's training setup [64], applying a confidence margin to all but the parallel examples.

We fine-tune an encoder-based model following the BGE-M3 architecture [6], adding a linear head to predict quality ratings. We choose BGE-M3 [6] for its strong multilingual representation ability and lightweight design, which make it well-suited for large-scale multilingual scoring. The resulting rater achieves over 93% accuracy on the validation set and 97% on the training set, demonstrating strong multilingual preference modeling. The effect of translation quality and implementation details, including tokenizer settings and hyperparameters, are provided in Appendix A.

## 4 Experiments

### 4.1 Experimental Setups

**Dataset construction.** We build on the deduplication and heuristic-filtering pipelines of FineWeb-2 [46] to assemble a large web-crawl corpus. It comprises 1.5 trillion English tokens plus 3 trillion

tokens across 17 additional languages (Arabic, Chinese, Dutch, French, German, Indonesian, Italian, Japanese, Korean, Portuguese, Russian, Spanish, Thai, Turkish, Vietnamese, Malay, Tagalog). We then apply MuRater and other baselines to assign quality scores to every document. Although scoring trillions of tokens is compute-intensive, it parallelizes efficiently across GPUs, and batching strategies reduce overhead in practice. Corpus statistics are detailed in Appendix B.1.

**Baselines.** For the English experiments, we train and evaluate the model using only English pairwise data, comparing it against several established data-quality raters: **QuRater**, which selects data based on educational value [64]; **AskLLM**, which follows the prompt design in [52] using Flan-T5-XXL [8]; the **FineWeb-Edu** Class ifier[3], trained on 450K LLaMA3-70B-Instruct [4] labels to identify educational content; and **DCLM**[5], a fastText classifier trained on high quality dataset to differentiate between informative and low-quality web content.

For the multilingual experiments, we extend the QuRater framework [64] to build **QuRater-M**, which employs GPT-4o to annotate multilingual pairs via relative preference, following the same training pipeline as its English counterpart. We further include datasets from **HPTL** [12] and **FineWeb-2** [46] for comprehensive comparison, and evaluate against **FineWeb2-HQ** [39]. In addition, we analyze the difference between **MuRater(E)** and **MuRater(M)**: **MuRater(M)** scores multilingual pairs that have been translated into English, whereas **MuRater(E)** starts from rated English data and translates it into multilingual pair and cross-lingual pair form. Most multilingual evaluations cover 17 languages plus English, while the experiment involving **FineWeb2-HQ** is restricted to 13 overlapping languages (Arabic, Chinese, Dutch, French, German, Indonesian, Italian, Japanese, Portuguese, Russian, Spanish, Turkish, and Vietnamese) to ensure fair comparison.

Finally, we include a **Uniform** baseline for both settings, which randomly samples 50% more data than the other methods, following the setup of QuRating [64]. Details are provided in Appendix B.2.

**Training Setup.** We train a randomly initialized language model based on the LLaMA architecture [17] for a single epoch over the training corpus, with data presented in a randomly shuffled order. For most experiments, the model comprises 1.2 billion parameters and employs a standard transformer architecture [60] augmented with rotary position embeddings (RoPE) [56]. To accommodate the multilingual setting, we extend the tokenizer vocabulary through retraining on the multilingual corpus. Comprehensive architectural and tokenizer details are provided in Appendix B.3.

Building on this setup, we construct the training corpora for both English and multilingual experiments. For the English setting, we select the top-scored 200 billion tokens from the full pool of 1.5 trillion tokens across all baseline methods and MuRater. In the multilingual setup, we apply all methods to score and select the top 10% of tokens within each language, yielding roughly 300 billion tokens in total. These multilingual tokens are then combined with the 200 billion English tokens to form a unified 500-billion-token pretraining corpus.

To further assess robustness and scalability under varied training conditions, we additionally conduct experiments with a 7B-parameter model sharing the same LLaMA architecture as the 1.2B model. This larger model is pretrained on 1T tokens, with 16.5% multilingual data selected by either MuRater or QuRater-M, while the remaining 83.5%—comprising English, code, and math data—remains identical across both setups. The multilingual portion follows the same language distribution as in the main experiments, enabling a more comprehensive evaluation of generalization across heterogeneous data sources.

**Evaluation Benchmarks.** We assess the performance of our pretrained models using the `lm-evaluation-harness` framework [14]. For the English-only evaluation, we consider a suite of ten tasks spanning multiple linguistic competencies. These include six reading comprehension benchmarks—ARC-Easy, ARC-Challenge [10], SciQ [63], LogiQA [37], TriviaAQ[26] and BoolQ [9]; four commonsense reasoning tasks—HellaSwag [70], PIQA [3], OpenBookQA [40] and Wino-Grande [53]; and two World Knowledge tasks—Natural Questions (NQ) [29] and MMLU [23]. For MMLU, we follow [2] and employ the `lighteval` variant to ensure more consistent and reliable comparisons.

---

[3] https://huggingface.co/HuggingFaceFW/fineweb-edu-classifier
[4] https://huggingface.co/meta-llama/Meta-Llama-3-70B-Instruct
[5] https://huggingface.co/mlfoundations/fasttext-oh-eli5

For the multilingual evaluation, we utilize translated versions of several English benchmarks [21] alongside multilingual-native datasets. Similar to English benchmarks, we divide the task into three categories, including reading comprehension, commonsense reasoning, and world knowledge understanding. Reading comprehension is evaluated using translated versions of ARC-Easy and ARC-Challenge [10], StoryCloze [41], and XNLI [11], which assess contextual understanding and inference. Commonsense reasoning is tested with HellaSwag [70], XCOPA [49], and XWinograd [58], focusing on event causality, semantic plausibility, and everyday reasoning across languages. World Knowledge evaluation includes MMLU [23], BMLAMA [50], and FLORES [16], which examine factual knowledge, translation quality, and multilingual alignment.

To strengthen the evaluation of language-specific knowledge, we also incorporate localized MMLU variants—CMMLU [32], VMLU[6], IndoMMLU [28], JMMLU[7], and AMMLU[8]—to construct a region-specific multilingual extension of MMLU, denoted as MMLU_L and add to the world knowledge category. Together, these benchmarks provide a comprehensive evaluation of cross-lingual comprehension, commonsense inference, and knowledge-grounded reasoning, enabling a holistic assessment of multilingual LLM performance. Detailed information of benchmark statistics and language coverage is provided in Appendix C.

## 4.2 Main Results

### 4.2.1 Multilingual Results

Table 1: Results on multilingual benchmarks with different training setups. Best results within each setting are shown in **bold**

| Selection Method | Reading Comprehension (5 tasks) | Commonsense Reasoning (2 tasks) | World Knowledge (4 tasks) | Average (11 tasks) |
|---|---|---|---|---|
| *18 Languages Results* | | | | |
| Uniform | 53.16 | 54.58 | 38.25 | 48.66 |
| HPLT-2 | 50.38 | 49.77 | 36.96 | 45.70 |
| FineWeb-2 | 50.83 | 52.48 | 35.53 | 46.28 |
| QuRater-M | 54.58 | 54.87 | 38.12 | 49.19 |
| MuRater(M) | 54.91 | 55.48 | 39.68 | 50.02 |
| MuRater(E) | **56.05** | **56.42** | **40.40** | **50.96** |
| *13 Languages Results* | | | | |
| FineWeb2-HQ | 53.05 | 55.54 | 38.31 | 48.97 |
| MuRater(E) | **55.95** | **58.30** | **41.17** | **51.81** |
| *7B Model Results* | | | | |
| QuRater-M | 61.96 | 63.28 | 43.31 | 56.18 |
| MuRater | **62.78** | **64.40** | **44.50** | **57.23** |

As shown in Table 1 and Figure 2, MuRater substantially outperforms existing multilingual baselines across nearly all evaluation categories and settings. Under the 18-language configuration, MuRater(E) achieves the highest category-averaged scores in all three categories, outperforming all baselines. These consistent gains highlight MuRater's capacity to identify high-quality, semantically rich, and educationally valuable text, even when faced with heterogeneous multilingual inputs. In particular, the large improvements in reasoning-oriented benchmarks (e.g., ARC and MMLU) suggest that MuRater selects examples with deeper conceptual structure and higher linguistic clarity, thereby enhancing model comprehension and reasoning generalization.

The performance gap between MuRater(M) and MuRater(E) further highlights the advantages of English-anchored training. MuRater(E) leverages English-rated pairs and projects the resulting preferences into the multilingual space, providing more stable and transferable supervision signals. We attribute this benefit to the broader topical and stylistic diversity of English corpora, which expose the rater to richer linguistic variation and more representative pairwise patterns during training. As a result, MuRater(E) learns to generalize beyond language corpus boundaries and more effectively

---

[6]https://vmlu.ai/
[7]https://huggingface.co/datasets/nlp-waseda/JMMLU
[8]https://huggingface.co/datasets/Hennara/ammlu

capture shared semantic dimensions across languages, yielding stronger cross-lingual alignment and greater robustness to translation-induced noise.

On the 13-language subset, MuRater(E) continues to outperform FineWeb2-HQ by roughly 3 points on average, achieving leading results across all three evaluation dimensions. When scaled to the 7B model trained on 1T tokens, MuRater maintains its superiority, reaching higher performance in all three respective categories—demonstrating consistent gains across both model sizes and data distributions. These results confirm that MuRater generalizes effectively to diverse linguistic environments and evaluation conditions, offering a more scalable and reliable framework for multilingual data quality estimation. Detailed results on each language are shown in Appendix D

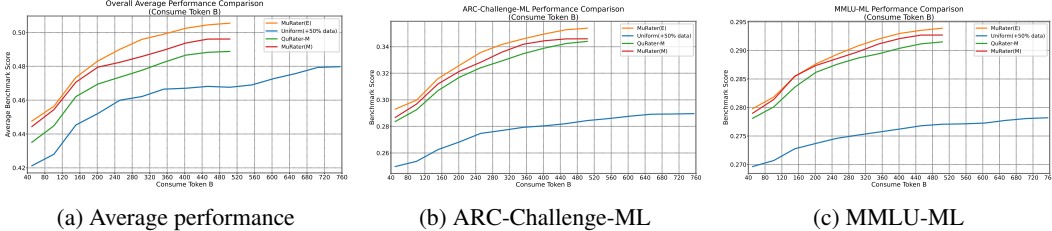

(a) Average performance          (b) ARC-Challenge-ML          (c) MMLU-ML

Figure 2: Performance of different selection methods on ARC-Challenge-ML, MMLU-ML, XWinograd, and the overall average across all tasks during training on 200B English + 300B multilingual tokens.

### 4.2.2 English-only Results

Table 2: Performance of different selection method over all different downstream tasks. Best results of each task category is marked in black. Detailed results are performed in Appendix D.

| Selection Method | Reading Comprehension (6 tasks) | Commonsense Reasoning (4 tasks) | World Knowledge (2 tasks) | Average (12 tasks) |
|---|---|---|---|---|
| Uniform (+50% data) | 43.93 | 59.06 | 20.36 | 48.70 |
| AskLLM | 42.83 | 58.40 | 20.21 | 47.82 |
| DCLM | 46.00 | 58.99 | 22.37 | 50.23 |
| FineWeb_Edu | 45.71 | 57.49 | 22.00 | 49.49 |
| QuRater | 43.54 | 58.58 | 20.47 | 48.33 |
| MuRater | **47.13** | **59.95** | **22.53** | **51.23** |

The results in Table 2 indicate that our proposed rater successfully consolidates the strengths of existing rating methodologies, leading to consistent improvements in pretrained model performance across all categories of evaluation tasks. Baseline comparisons reveal that each selection method exhibits distinct preferences for data, which translate into varying levels of effectiveness on different downstream tasks. For instance, as shown in Figure 3, DCLM yields strong results on HellaSwag but underperforms on ARC-Challenge. Conversely, QuRater achieves competitive performance on ARC-Challenge but demonstrates poor results on TriviaQA. In contrast, our MuRater integrates the advantages of these methods and achieves robust performance across nearly all benchmarks, outperforming other data selection baselines by margins ranging from 1 to 3.4 percent. The model trained with our rater consistently achieves superior results on all tasks throughout the training process. This indicates more stable and efficient learning, further validating the effectiveness of our data selection approach in enhancing the quality of LLM pretraining.

### 4.3 Ablation Study

### 4.3.1 Effectiveness of Cross-Lingual and Parallel Pair Integration

Incorporating cross-lingual pairs and parallel translations during training significantly improves the consistency of quality scoring across languages. To validate this, we assess multilingual raters on parallel corpora—semantically equivalent texts in different languages. As shown in Figure 4, our alignment-based training produces models with lower mean squared error (MSE) and slopes closer to one, indicating stronger cross-lingual consistency. In an ideal case, a language-agnostic rater would

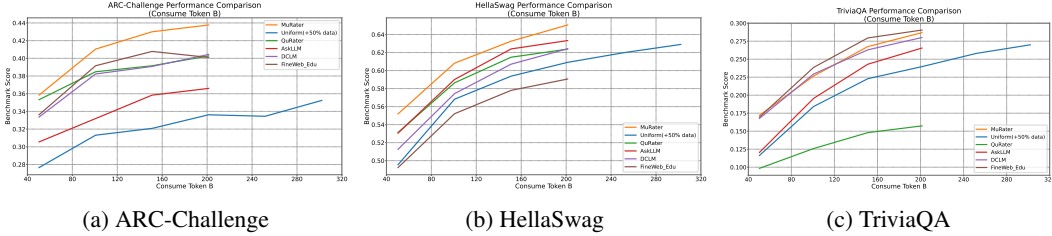

| (a) ARC-Challenge | (b) HellaSwag | (c) TriviaQA |

Figure 3: Performance of different selection methods on ARC-Challenge, HellaSwag, TriviaQA, and the overall average across 12 tasks during training on 200B English tokens

assign identical scores to parallel texts across languages, resulting in a slope of one and minimal MSE between their score sequences.

These findings highlight the importance of modeling interlingual relationships. By leveraging cross-language comparisons and parallel data, the rater learns language-invariant quality standards, enabling more reliable multilingual evaluation. A qualitative case study in Appendix E further supports this, showing that high-rated texts consistently exhibit greater fluency, coherence, and instructional value across languages. We further examine how the selected data is distributed across semantic domains in different languages. Detailed results are provided in Appendix B.1.

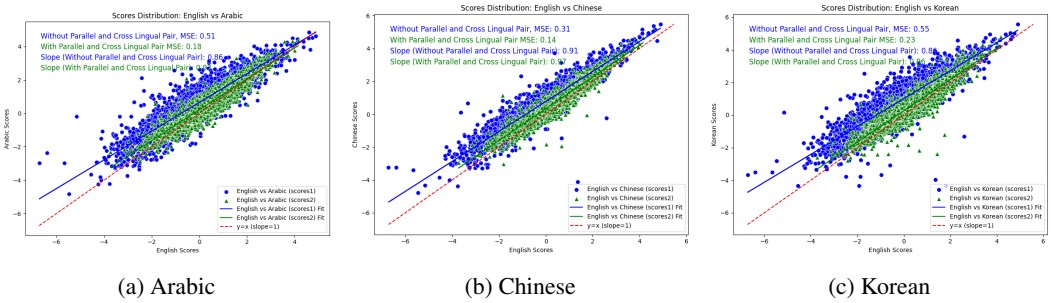

| (a) Arabic | (b) Chinese | (c) Korean |

Figure 4: Scatter plots of scores assigned by multilingual raters to 10,000 parallel documents across various languages. Green points represent ratings from raters trained with alignment using parallel and cross-lingual pairs, while blue points indicate scores from unaligned raters.

### 4.3.2 Comparison Between Pairwise and Pointwise Score Transfer

We examine the relative effectiveness of pairwise versus pointwise judgment methods for transferring English scoring capabilities to multilingual settings. Based on the translation quality evaluations, we select two high-performing languages, Arabic and Spanish, for the study. Specifically, we translate 200 English text pairs into Arabic and 200 pairs into Spanish. Each dataset is then annotated by GPT-4o using both pairwise and pointwise scoring strategies. For pointwise annotation, GPT-4o assigns quality scores on a 1–10 scale. The scoring prompts of both methods explicitly instruct GPT-4o to evaluate based on content quality alone, irrespective of language and are detailed in Appendix A.6. Each text or pair is scored 20 times, and the average is used as the final score. Given identical content across different languages, the ideal scenario is that a consistent model and prompt should yield nearly identical scores, regardless of the language and score strategies.

As shown in Figure 5, pointwise scores exhibit considerable variability across languages, particularly in the mid-quality range (scores between 3 and 6), despite relatively stable assessments at the high and low ends. Ideally, all points should align closely with the $y = x$ (slope = 1) line, which would indicate identical score for semantically equivalent content across languages. In contrast, pairwise judgments display strong cross-lingual consistency, with only minor deviations from this ideal alignment. These observations suggest that while translation quality is generally adequate, subtle translation biases can still influence absolute (pointwise) ratings. The pairwise approach, however, demonstrates greater robustness to such variation, underscoring its effectiveness for reliably transferring English scoring behavior to multilingual contexts.

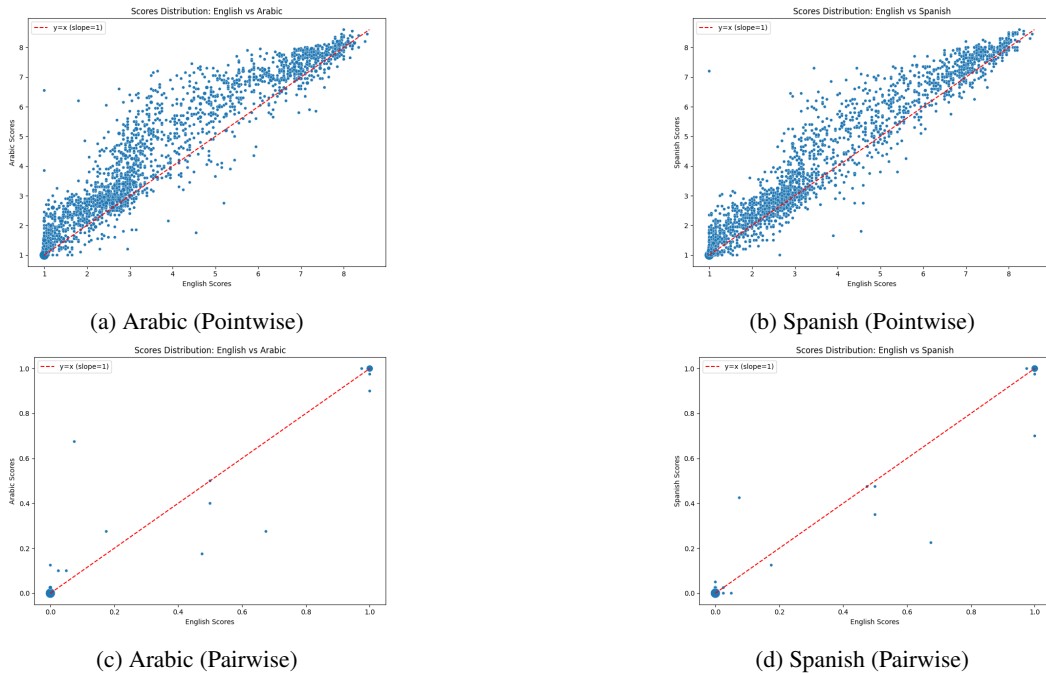

Figure 5: Scatter plots of average scores assigned by GPT-4o to Arabic and Spanish parallel data. Each point represents an average of 20 evaluations. Left: pointwise scoring. Right: pairwise scoring.

# 5    Conclusion

We introduced MuRating, a scalable multilingual data selection framework that aggregates multiple English raters via a Bradley–Terry pairwise model and transfers these judgments through translation to train a single multilingual MuRater over monolingual, cross-lingual, and parallel pairs. Applied to large web corpora, MuRater is used to pretrain both 1.2B- and 7B-parameter LLaMA-architecture models and delivers consistent improvements over strong baselines (QuRater, FineWeb2-HQ, AskLLM, DCLM) on multiple English and multilingual benchmarks. Ablation results further demonstrate the effectiveness of incorporating cross-lingual and parallel pairs, and confirm that pairwise supervision provides more stable multilingual scoring than pointwise methods. Analyses across translation fidelity and data composition, together with results at two model scales, indicate that MuRating is effective and scalable for large-scale multilingual data curation and yields reliable performance gains across evaluation settings.

## Limitations

Our current study focuses on 17 target languages excluding English, leaving substantial room for broader linguistic coverage. The reliance on GPT-4o introduces potential biases and idiosyncrasies inherent to proprietary large language models. Moreover, since the English raters used in our framework primarily focused on factual and informational content, our auto-rater exhibits limited performance on narrative and creative domains. While the proposed approach performs well on language-specific benchmarks, further research could explore language-specific rater designs or culturally aligned data selection strategies to better capture the unique characteristics of each language. Future work will also aim to incorporate higher-quality translations, expand to a wider range of languages, and develop adaptive data sampling techniques to enrich the diversity and representativeness of the curated multilingual corpus.

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

# A  Details of MuRater Model

## A.1  Different Annotation Method

**GPT annotation** We adopt the educational value prompt criteria from QuRating [64] as our annotation prompt for GPT-4o-08-06, as detailed below. This prompt is used to annotate a total of 300,000 document pairs. For each pair, we randomly extract a segment of $n$ tokens—based on the LLaMA tokenizer [59]—where $n$ is sampled from a uniform distribution $n \sim \text{Uniform}[256, 512]$ in 50% of cases, and fixed at 512 tokens otherwise. Annotation involves generating 20 predictions of either "A" or "B" per criterion and document pair (in either order). The total cost of dataset creation amounts to \$9,740.

---

**Pairwise Educational Value Prompt**

Compare two text excerpts and choose the text which has more educational value, e.g., it includes clear explanations, step-by-step reasoning, or questions and answers.
Aspects that should NOT influence your judgement:
1. Which language the text is written in
2. The length of the text
3. The order in which the texts are presented
Note that the texts are cut off, so you have to infer their contexts. The texts might have similar quality, but you should still make a relative judgement and choose the label of the preferred text.
[Option label a] ... text a ...
[Option label b] ... text b ...
Now you have to choose between either label a or label b. Respond only with a single word.

---

**AskLLM** We adopt the approach from [52] and use the following prompt to query Flan-T5-xxl [8] for annotation 300,000 document pairs.

---

**Ask-LLM prompt**

### This is a pretraining . . . . datapoint. ###
Does the previous paragraph demarcated within ### and ### contain informative signal for pre-training a large-language model? An informative datapoint should be well-formatted, contain some usable knowledge of the world, and strictly NOT have any harmful, racist, sexist, etc. content.
OPTIONS:
- yes
- no

---

**Fineweb and DCLM** For these two data selection methods, we directly use the open-sourced model to annotate documents to obtain the scores.

## A.2  Training details of MuRater and training accuracy

We adopt the XLM-RoBERTa architecture encoder model BGE-M3 [6] as the foundation of our multilingual rating model, MuRater, and fine-tune it by appending a linear regression head to the transformer output to predict quality scores. The fine-tuning process employs a confidence margin threshold of 50%, defined as $\epsilon = |p_A - p_B| = |2p_{B \succ A} - 1|$ for a prediction between text pairs $(t_A, t_B)$ [64]. Fine-tuning is conducted over 3 epochs with a batch size of 512 and a learning rate of $2 \times 10^{-5}$. We set $\lambda$ to 0.5. Performance on held-in and held-out sets is summarized in Table 3. Notably, BGE-M3 supports over 100 languages and leverages large-scale multilingual unsupervised data to learn a shared semantic space, making it particularly effective for multilingual and cross-lingual retrieval and rating tasks.

## A.3  Translation

The translation prompts is

| Evaluation Dataset | Confidence Margin | Accuracy |
|---|---|---|
| Training set (held-in) | 50% | 94.3% |
| | 80% | 97.2% |
| Validation set (held-out) | 50% | 90.7% |
| | 80% | 93.1% |

Table 3: Prediction accuracy of MuRater on held-in and held-out datasets under different confidence margins.

---

**Translation Prompt**

Please translate the following {lang} text into {lang2}. Your translations must convey all the content in the original text and cannot involve explanations or other unnecessary information. Please ensure that the translated text is natural for native speakers with correct grammar and proper word choices.
Your translation must also use exact terminology to provide accurate information even for the experts in the related fields.
The text is : {text}

---

We translate a total of 600,000 English document pairs evenly across 17 languages using the GPT-4o-08-06 model, with the overall translation cost amounting to $18,720.

## A.4 Human translation quality evaluation

To evaluate the translation quality of GPT-4o outputs, we employed professional human translators to assess a selected subset of the generated texts. All evaluators possessed CEFR C1-level or higher proficiency in both English and the respective target language. Each language translation was reviewed by a single expert. Evaluators were compensated at a rate of $16 per hour, with each assessment session lasting approximately 4 hours. the annotation criteria for translation quality is shown below.

---

**Annotation Criteria**

5 points: The translation accurately reflects the meaning of the original text, is fluent, and contains no errors.
4 points: The translation generally reflects the meaning of the original text, with most sentences being fluent, but there are slight inaccuracies in the use of non-key terms or non-idiomatic phrases.
3 points: The translation conveys the general idea of the original text, but contains significant errors such as improper translation of key terms, incorrect word order, omissions, mistranslations, or untranslated segments.
2 points: The translation is largely incomprehensible or unfaithful to the original text, with serious errors including issues of order, logic, or severe grammatical mistakes.
1 point: The translation is completely incomprehensible or entirely unfaithful to the original text, or it fails to convey the original meaning entirely, being obscure and difficult to understand.
Please note that all sentences are excerpts from web content, so the last sentence of each segment, which may be unclear, is not considered in the evaluation.

---

We ensured adherence to ethical standards in our human annotation process:

- **Fair Compensation**: All annotators received compensation at or above the minimum wage standards of their respective regions.

- **Informed Consent**: Annotators were provided with clear instructions and information about the annotation tasks. Participation was voluntary, and informed consent was obtained prior to their involvement.

- **Institutional Review**: Our study underwent review and received approval from the Institutional Review Board (IRB) at our institution, ensuring that the research met ethical standards for studies involving human participants.

- **Transparency**: Detailed information regarding the annotation are included in the supplementary materials to promote transparency and reproducibility.

## A.5   Translation Quality Assement

We assess the quality of our translations through human evaluation. Expert annotators are provided with 50 pairs of source and translated texts and asked to rate translation quality on a scale mentioned above. As shown in Figure 6, the overall translation quality of GPT-4o is high, with most languages achieving average scores above 4. Notably, performance on Japanese and Thai is comparatively lower, though still above 3.5, suggesting acceptable translation quality for these languages.

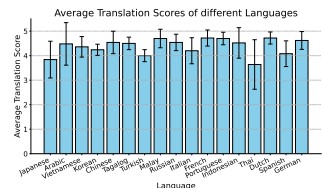

Figure 6: Translation average scores of various languages.

To further evaluate the robustness of our framework with respect to translation quality, we conducted a supplementary experiment using the open-sourced LLM Qwen 3-8B [57]. Qwen 3-8B achieves a FLORES chrF score of 56, which is lower than GPT-4o's score of 62 for the same language set. We employed both Qwen 3-8B and GPT-4o to translate our training preference pairs and trained separate MuRater models based on each translation. Each MuRater was then applied to score 10,000 multilingual documents per language. We compared the annotation outputs between the two models using Pearson correlation and Kendall's Tau.

Table 4: Agreement between MuRater models trained with translations from Qwen 3-8B vs. GPT-4o, measured by Kendall's Tau and Pearson correlation.

| **Language** | ja | de | es | ar | id | pt | th | fr | vi |
|---|---|---|---|---|---|---|---|---|---|
| Kendall's Tau | 0.8930 | 0.9005 | 0.8951 | 0.8645 | 0.8903 | 0.8964 | 0.8834 | 0.8924 | 0.8867 |
| Pearson Corr. | 0.9835 | 0.9854 | 0.9843 | 0.9793 | 0.9822 | 0.9848 | 0.9803 | 0.9831 | 0.9825 |

| **Language** | it | ko | ms | tl | ru | tr | zh | nl | |
|---|---|---|---|---|---|---|---|---|---|
| Kendall's Tau | 0.8851 | 0.8885 | 0.8427 | 0.8749 | 0.8942 | 0.8822 | 0.9044 | 0.8991 | |
| Pearson Corr. | 0.9815 | 0.9821 | 0.9644 | 0.9755 | 0.9850 | 0.9818 | 0.9861 | 0.9843 | |

As shown in Table 4, both metrics indicate consistently high agreement across all evaluated languages. These findings suggest that even when relying on a weaker translation model such as Qwen 3-8B, MuRater can still be effectively trained, provided that the relative preference information is preserved. This demonstrates the robustness of our approach to moderate variations in translation quality.

## A.6   Pointwise Score

The pointwise scoring prompt is provided below. We instruct GPT-4o to evaluate each text 10 times, then compute the average of these scores to determine the final rating. The scoring range is from `grade_min = 1` to `grade_max = 10`.

> **Pointwise prompt evaluation for educational value**
>
> I need to rate a text excerpt on a scale of {grade_min} to {grade_max} (inclusive) based on its educational value, e.g., it includes clear explanations, step-by-step reasoning, or questions and answers.
> Aspects that should NOT influence your judgement: 1. Which language the text is written in 2. The length of the text
> Note that the text is cut off, so you have to infer its context.
> [Text] ... {text} ...
> Now assign a number grade between {grade_min} to {grade_max} (inclusive). Respond only with a single digit. The score for the quality of the text is:

# B  Experiment Setup Details

## B.1  Dataset

We use the 16 recent snapshots from FineWeb-2 as our raw data before MuRater and other baselines annotation, namely `CC-MAIN-2021-39`, `2021-43`, `2021-49`, `2022-05`, `2022-21`, `2022-27`, `2022-33`, `2022-40`, `2022-49`, `2023-06`, `2023-14`, `2023-23`, `2023-40`, `2023-50`, `2024-10`, and `2024-18`.

To analyze domain composition, we employ NVIDIA's multilingual domain classifier[9] to annotate the domain distribution of our dataset. Figures 7–11 illustrate the domain shifts before and after applying the MURATER-based selection. The results show that MURATER systematically prioritizes World Knowledge domains such as *People and Society*, *Health*, and *Science*, which are typically well-structured and rich in informational content—properties particularly beneficial for large language model pretraining. However, the resulting domain distributions vary across languages, primarily reflecting intrinsic differences in the domain composition of their respective source corpora.

We adopt the open-sourced FastText language identification model [27], which supports 176 languages and is widely used in large-scale multilingual data pipelines. For multilingual data selection, we retain the top 10% of highest-scoring documents per language from the raw corpus, preserving the natural data composition of Common Crawl and maintaining a distribution similar to full set of FineWeb-2. The initial token ratio across languages is: `ru (14.29%)`, `es (12.83%)`, `ja (12.19%)`, `de (12.19%)`, `zh (9.26%)`, `fr (8.98%)`, `it (7.29%)`, `pt (4.58%)`, `nl (4.53%)`, `vi (3.21%)`, `id (2.88%)`, `ar (2.75%)`, `tr (2.20%)`, `th (1.51%)`, `ko (1.41%)`, `tl (0.04%)`, `ms (0.02%)`.

We then apply temperature-based sampling with $\tau = 3.33$, following standard multilingual pretraining practices [69]. The final token ratios used in training are: `ru (8.08%)`, `es (7.78%)`, `ja (7.62%)`, `de (7.62%)`, `zh (7.00%)`, `fr (6.93%)`, `it (6.50%)`, `pt (5.71%)`, `nl (5.70%)`, `vi (5.07%)`, `id (4.86%)`, `ar (4.78%)`, `tr (4.52%)`, `th (4.15%)`, `ko (4.11%)`, `tl (2.75%)`, `ms (2.62%)`.

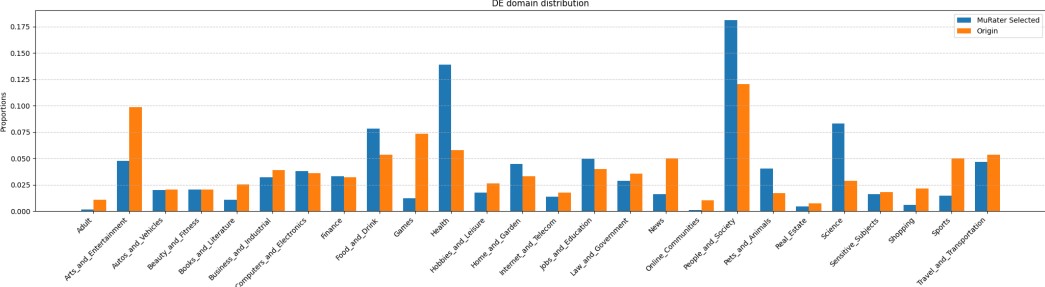

Figure 7: Domain distribution of German corpus

---

[9] https://huggingface.co/nvidia/multilingual-domain-classifier

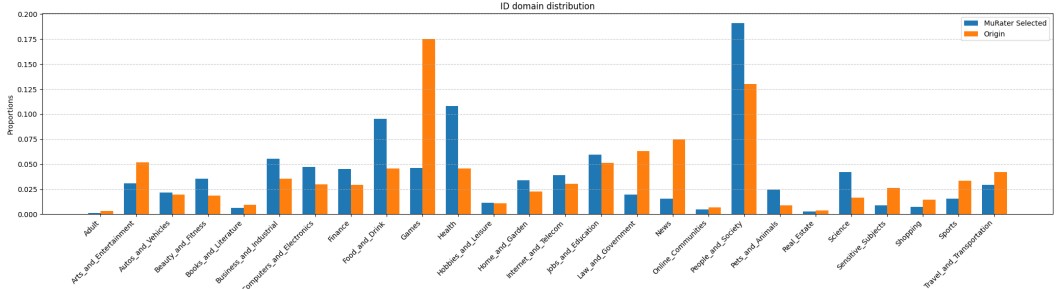

Figure 8: Domain distribution of France corpus

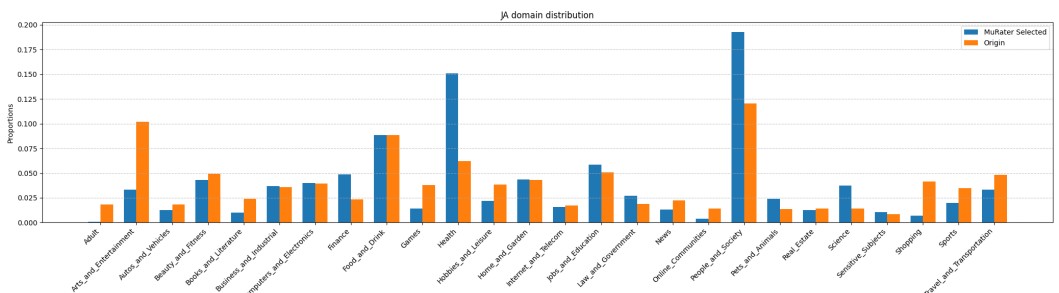

Figure 9: Domain distribution of Japanese corpus

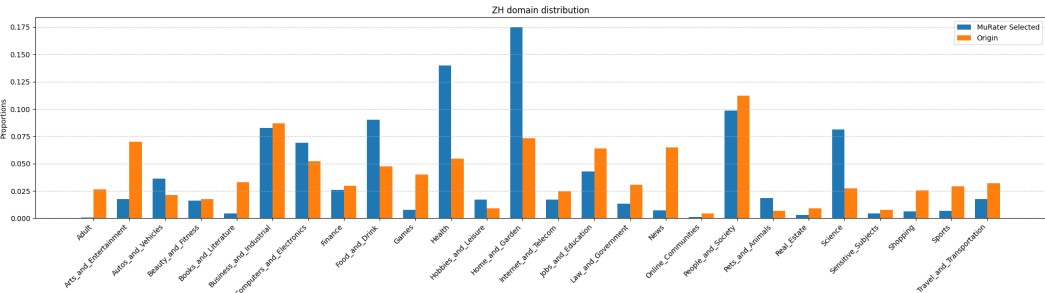

Figure 10: Domain distribution of Chinese corpus

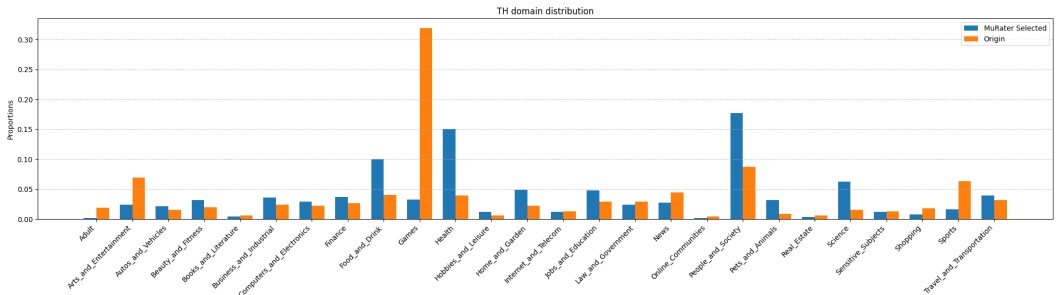

Figure 11: Domain distribution of Thai corpus

| Model configuration | Values |
|---|---|
| Attention head | 16 |
| Layers | 24 |
| Hiddent size | 2048 |
| Intermediate layer dimension | 5504 |
| maximum position embedding | 4096 |
| layer normalization epsilon | $1 \times 10^{-5}$ |
| **Training Hyperparameters** | **Values** |
| Batch size | 3072 |
| Sequence length | 4096 |
| Optimizer | AdamW |
| Learning rate | $4.3 \times 10^{-4}$ |
| Learning rate schedule | Cosine decay to 10% of inital value |
| Traning steps | Varied based on the total token budget |
| Precision | bf16(mxied-precision training) |

Table 5: Model configuration and Training Hyperparameters for pretraining LLms

## B.2 Baselines

We follow the same annotation procedure for the English datasets of QuRater, AskLLM, DCLM, and FineWeb-Edu as described in Appendix A. For QuRater-M, we apply the same prompting approach (also detailed in Appendix A) and instruct GPT-4o to annotate 300,000 multilingual pairs, focusing exclusively on content regardless of the language. We then fine-tune the multilingual QuRater baseline using both English and multilingual data, leveraging the BGE-M3 model [6] and the identical training hyperparameters outlined in Appendix A.

## B.3 Model Architecture

We utilize a transformer architecture based on the LLaMA-2 model [59], configured to contain approximately 1.2 billion parameters. Models are randomly initialized before pretraining. The detailed information for the model configuration and training hyperparamters is shown in Table 5. We preprocess our training corpus to train a custom Byte-Pair Encoding (BPE) tokenizer using the BBPE algorithm, yielding a vocabulary of 250,000 tokens for use in our training experiments. The main experiments is conducted using 64 NVIDIA H100 GPUs, with an average runtime of approximately 70 hours per experiment.

## C Evaluation Benchmarks

All task evaluations are conducted using the `lm-evaluation-harness` framework [14]. For English in-context learning tasks, we use the following benchmakrs:

- **ARC-Easy and ARC-Challenge** [10] (25-shot): Multiple-choice science questions from grade school exams, assessing models' ability to apply scientific knowledge and reasoning.
- **SciQ** [63] (0-shot): Crowdsourced multiple-choice science questions covering physics, chemistry, and biology, designed to evaluate scientific understanding.
- **LogiQA** [37] (0-shot): Logical reasoning questions derived from Chinese civil service exams, testing deductive reasoning capabilities.
- **TriviaQA** [26] (5-shot): Reading comprehension dataset with question-answer pairs authored by trivia enthusiasts, accompanied by evidence documents.
- **BoolQ** [9] (5-shot): Yes/no questions with associated passages, evaluating models' ability to answer naturally occurring questions.

For commonsense reasoning, we evaluate on:

- **HellaSwag** [70] (10-shot): Sentence completion tasks requiring commonsense inference to select the most plausible continuation.

- **PIQA** [3] (5-shot): Physical commonsense reasoning questions, focusing on everyday tasks and interactions.
- **OpenBookQA** [40] (10-shot): Multiple-choice questions based on elementary science facts, requiring both factual knowledge and reasoning.
- **WinoGrande** [53] (5-shot): Pronoun resolution tasks designed to test commonsense reasoning at scale.

Additionally, two World Knowledge tasks are evaluated:

- **Natural Questions (NQ)** [29] (5-shot): Real user questions paired with answers from Wikipedia, assessing open-domain question answering.
- **MMLU** [23] (5-shot): A benchmark covering 57 subjects across various domains, measuring multitask language understanding.

For evaluating translated benchmarks, we use the MuBench dataset [21] and conduct evaluations across 18 languages present in our training set. In the multilingual setting, we evaluate:

- **ARC-Easy and ARC-Challenge** (25-shot): Translated versions of the science question benchmarks, assessing cross-lingual reasoning.
- **HellaSwag** (10-shot): Evaluating commonsense reasoning in multiple languages through sentence completion tasks.
- **MMLU** (5-shot): Multilingual evaluation of multitask language understanding across diverse subjects.
- **StoryCloze** [41] (0-shot): Narrative understanding task where models choose the correct ending to a four-sentence story.
- **BMLAMA** [50] (0-shot): Multilingual factual knowledge probing dataset, assessing cross-lingual consistency in language models.
- **XCOPA** [49] (5-shot): Causal commonsense reasoning tasks translated into multiple languages, evaluating cross-lingual inference.
- **XNLI** [11] (5-shot): Cross-lingual natural language inference benchmark, testing entailment and contradiction detection.
- **XWinograd** [58] (5-shot): A multilingual benchmark for evaluating localized knowledge and reasoning abilities of large language models across diverse languages.
- **MultiLoKo** [24] (5-shot): A large-scale multilingual evaluation suite designed to assess factual knowledge, reasoning, and question answering across 45 languages, emphasizing both cross-lingual consistency and language-specific understanding.
- **FLORES** [16] (5-shot): Multilingual machine translation benchmark, evaluating translation quality across diverse languages.
- **MMLU_L** (5-shot): A localized version of MMLU, focusing on both general knowledge and language-specific knowledge and reasoning tasks.

## D    Detailed Results

### D.1    English Detailed Results

Table 6 presents the detailed performance of various selection methods across individual downstream tasks. Our method consistently outperforms others on most tasks, with notable improvements on ARC, HellaSwag, and MMLU.

### D.2    Multilingual Detailed Results

Detailed results of different data-selection methods across individual downstream benchmarks. For the 7B experiments, we additionally include the MultiLoKo benchmark [24] to assess cultural and regional knowledge across languages. Since its scores were too low to be meaningful under the 1.2B training setup, we do not report the results here.

We also display the detailed results of each benchmark and each language below.

Table 6: Detailed performance of differnt selection method over all downstream tasks with all values in percentages and per-benchmark maximum highlighted in bold.

| Data Selection Method | ARC_Challenge | ARC_Easy | BoolQ | HellaSwag | LogiQA | MMLU | NQ | OpenBookQA | PIQA | TriviaQA | WinoGrande | SciQ | Average |
|---|---|---|---|---|---|---|---|---|---|---|---|---|---|
| Uniform (+*50% data*) | 35.24 | 66.50 | 64.46 | 62.90 | 28.88 | 32.85 | **7.87** | 37.00 | 75.73 | 27.00 | **60.62** | 85.40 | 48.70 |
| Askllm | 36.60 | 67.63 | 59.76 | 63.33 | 26.57 | 32.89 | 7.53 | 35.60 | 76.82 | 26.55 | 57.85 | 82.70 | 47.82 |
| DCLM | 40.44 | 73.78 | 64.07 | 62.42 | 28.73 | 35.42 | 9.31 | 37.40 | 76.06 | 28.01 | 60.06 | 87.00 | 50.23 |
| FineWeb_Edu | 40.10 | 72.39 | **64.62** | 59.06 | 26.88 | 36.01 | 7.98 | **38.20** | 74.27 | **29.05** | 58.41 | 86.90 | 49.49 |
| QuRater | 40.27 | 72.14 | 61.93 | 62.38 | 28.88 | 35.26 | 5.68 | 38.60 | 75.63 | 15.74 | 57.70 | 85.80 | 48.33 |
| MuRater | **43.77** | **75.84** | 64.28 | **65.06** | **30.11** | **37.24** | 7.81 | **38.20** | **77.04** | 28.69 | 59.51 | **87.20** | **51.23** |

Table 7: Performance of different data-selection strategies across downstream tasks when mixing 200B English and 300B multilingual tokens. **MuRater(M)** denotes scoring multilingual pairs translated into English, while **MuRater(E)** uses rated English data translated into multilingual pair form. Best results within each setting are shown in **bold**.

| Selection Method | MMLU (L) | ARC C_ML | ARC E_ML | FLORES | Hella swag_ML | MMLU (T) | XCOPA | XNLI | Story Cloze_ML | XWino | BMLAMA | Average |
|---|---|---|---|---|---|---|---|---|---|---|---|---|
| *18 Languages* | | | | | | | | | | | | |
| Uniform | 29.98 | 28.74 | 49.03 | 46.66 | 44.56 | 27.83 | 64.60 | 42.33 | **69.28** | **76.40** | 48.55 | 48.00 |
| HPLT-2 | 29.24 | 26.94 | 45.95 | 42.66 | 39.87 | 27.58 | 59.67 | 42.14 | 65.09 | 71.77 | 48.38 | 45.39 |
| FineWeb-2 | 28.33 | 27.50 | 45.97 | 44.52 | 42.39 | 27.75 | 62.57 | 41.00 | 66.77 | 72.89 | 41.51 | 45.56 |
| QuRater-M | 30.86 | 33.89 | 57.53 | 46.60 | 46.77 | 29.18 | 62.97 | 44.39 | 65.86 | 71.21 | 45.84 | 48.65 |
| MuRater(M) | 31.86 | 34.26 | 58.21 | **47.43** | 46.54 | 29.28 | 64.43 | 42.07 | 67.08 | 72.92 | 50.13 | 49.47 |
| MuRater(E) | **31.96** | **35.01** | **58.98** | 47.27 | **47.11** | **29.40** | **65.73** | **44.46** | 67.67 | 74.13 | **52.97** | **50.43** |
| *13 Languages Subset* | | | | | | | | | | | | |
| FineWeb2-HQ | 30.52 | 30.91 | 54.37 | **50.97** | 46.15 | 28.66 | 64.92 | 41.79 | **69.48** | 68.72 | 43.08 | 48.14 |
| MuRater(E) | **31.96** | **36.03** | **61.63** | 50.11 | **49.17** | **29.43** | **67.44** | **44.49** | 68.75 | **68.83** | **53.19** | **51.00** |

Table 8: Comparison of MuRater and QuRater-M when training a 7B model on 1T tokens with 16.5% multilingual data.

| | MMLU_L | ARC_C_ML | ARC_E_ML | FLORES | HSWAG | MMLU_T | XCOPA | XNLI | S. Cloze | XWino | BMLAMA | MultiLoKo | Avg. |
|---|---|---|---|---|---|---|---|---|---|---|---|---|---|
| QuRater-M | 35.93 | 42.36 | 65.20 | 55.15 | 57.29 | 32.54 | 69.27 | **45.13** | 74.50 | 82.60 | 49.61 | 8.87 | 51.54 |
| MuRater | **36.87** | **43.38** | **66.87** | **55.38** | **57.76** | **32.93** | **71.03** | 45.06 | **75.42** | **83.19** | **52.82** | **10.61** | **52.61** |

Table 9: Detailed per-language performance on across **ARC-Easy**. Bold indicates the best result for each language.

| Method | AR | DE | EN | ES | FR | ID | IT | JA | KO | MS | NL | PT | RU | TA | TH | TR | VI | ZH |
|---|---|---|---|---|---|---|---|---|---|---|---|---|---|---|---|---|---|---|
| Uniform | 42.21 | 53.07 | 65.57 | 56.40 | 53.66 | 52.02 | 52.86 | 49.07 | 44.44 | 45.62 | 51.43 | 54.17 | 50.67 | 37.88 | 39.44 | 46.72 | 48.44 | 55.47 |
| QuRater-M | 52.69 | 62.25 | 72.94 | 65.74 | 63.59 | 61.32 | 62.71 | 57.62 | 53.58 | 54.29 | 60.44 | 63.51 | 59.34 | 42.85 | 43.90 | 55.72 | 54.67 | 63.72 |
| MuRater(M) | 52.19 | 62.79 | 72.85 | 66.54 | 63.85 | 63.30 | 62.58 | 58.00 | 53.96 | 55.89 | **61.70** | 63.47 | 59.85 | 43.35 | **45.75** | 56.40 | 56.19 | 63.76 |
| MuRater(E) | **52.82** | **63.22** | **73.91** | **67.55** | **63.97** | **63.68** | **63.80** | **58.88** | **54.59** | **56.78** | 61.62 | **65.24** | **60.98** | **44.53** | 45.50 | **57.28** | **57.03** | **65.11** |

Table 10: Detailed per-language performance on across **ARC-Challenge**. Bold indicates the best result for each language.

| Method | AR | DE | EN | ES | FR | ID | IT | JA | KO | MS | NL | PT | RU | TA | TH | TR | VI | ZH |
|---|---|---|---|---|---|---|---|---|---|---|---|---|---|---|---|---|---|---|
| Uniform | 27.82 | 30.03 | 32.25 | 30.12 | 30.03 | 28.92 | 30.03 | 28.92 | 26.54 | 29.01 | 28.33 | 30.55 | 29.61 | 24.83 | 28.24 | 28.16 | 28.58 | 28.92 |
| QuRater-M | 30.55 | 35.58 | 41.89 | 36.95 | 35.92 | 34.90 | 37.20 | 33.87 | 32.59 | 34.56 | 34.13 | 36.60 | 35.32 | **28.16** | 28.75 | 32.34 | 32.59 | 36.18 |
| MuRater(M) | 30.20 | **36.95** | 41.13 | **39.25** | 35.75 | 35.07 | 35.49 | 34.39 | 33.36 | 32.51 | 34.56 | 37.71 | 35.84 | 27.99 | 29.78 | 33.45 | **33.70** | 36.35 |
| MuRater(E) | **31.91** | 36.09 | **42.06** | 39.08 | **37.29** | 36.18 | **38.57** | 35.67 | 33.45 | 36.01 | 34.81 | 39.25 | 37.20 | 27.13 | 30.20 | 35.07 | 31.48 | 35.84 |

Table 11: Detailed per-language performance on across **HellaSwag**. Bold indicates the best result for each language.

| Method | AR | DE | EN | ES | FR | ID | IT | JA | KO | MS | NL | PT | RU | TA | TH | TR | VI | ZH |
|---|---|---|---|---|---|---|---|---|---|---|---|---|---|---|---|---|---|---|
| Uniform | 39.52 | 47.50 | 60.48 | 51.46 | 51.37 | 47.01 | 49.52 | 42.09 | 38.53 | 43.34 | 48.36 | 50.17 | 45.76 | 36.68 | 35.62 | 40.05 | 43.95 | 46.59 |
| QuRater-M | 41.83 | 49.71 | 61.61 | 54.05 | 54.48 | 49.95 | 51.87 | 43.88 | **40.63** | 45.10 | 50.20 | 52.71 | **48.92** | 37.93 | 37.46 | 42.53 | **46.33** | 47.50 |
| MuRater(M) | 41.65 | 49.62 | **62.46** | 54.00 | 54.62 | 49.94 | 51.89 | 43.53 | 40.32 | 45.60 | 50.08 | 52.63 | 48.03 | 37.41 | 37.10 | 42.15 | 45.33 | 47.23 |
| MuRater(E) | **42.17** | **50.23** | 62.30 | **54.84** | **55.13** | 50.36 | **52.13** | **44.39** | 40.48 | **46.16** | **50.89** | **53.55** | 48.53 | 37.69 | 37.30 | **42.62** | 46.28 | **48.06** |

Table 12: Detailed per-language performance on across **MMLU**. Bold indicates the best result for each language.

| Method | AR | DE | EN | ES | FR | ID | IT | JA | KO | MS | NL | PT | RU | TH | TL | TR | VI | ZH |
|---|---|---|---|---|---|---|---|---|---|---|---|---|---|---|---|---|---|---|
| Uniform | 0.2628 | 0.2769 | 0.2968 | 0.2782 | 0.2810 | 0.2787 | 0.2741 | 0.2777 | 0.2748 | 0.2791 | 0.2772 | 0.2817 | 0.2708 | 0.2701 | 0.2743 | 0.2742 | 0.2799 | 0.2824 |
| QuRater-M | 0.2747 | 0.2949 | 0.3180 | 0.2935 | 0.2975 | 0.2988 | 0.2915 | 0.2911 | 0.2852 | 0.2880 | 0.2979 | 0.2953 | 0.2893 | **0.2812** | 0.2821 | 0.2872 | 0.2915 | 0.2947 |
| MuRater(M) | 0.2727 | 0.2957 | 0.3235 | 0.2908 | 0.3018 | **0.3000** | 0.2944 | 0.2909 | **0.2919** | 0.2877 | 0.2968 | 0.2997 | 0.2907 | 0.2797 | 0.2812 | 0.2874 | 0.2944 | 0.2914 |
| MuRater(E) | **0.2765** | **0.3033** | 0.3206 | **0.2983** | 0.3010 | 0.2989 | 0.2905 | **0.2936** | 0.2871 | **0.2925** | **0.2976** | 0.2988 | 0.2886 | 0.2813 | **0.2850** | 0.2868 | **0.2967** | 0.2949 |

Table 13: Detailed per-language performance on across **StoryCloze**. Bold indicates the best result for each language.

| Method | AR | DE | EN | ES | FR | ID | IT | JA | KO | MS | NL | PT | RU | TH | TL | TR | VI | ZH |
|---|---|---|---|---|---|---|---|---|---|---|---|---|---|---|---|---|---|---|
| Uniform | 0.6161 | **0.7237** | **0.7570** | 0.7221 | **0.7237** | 0.6974 | **0.6950** | 0.6718 | **0.6463** | 0.6881 | **0.7074** | 0.7214 | 0.7090 | 0.6502 | **0.5967** | 0.6238 | 0.6865 | 0.7136 |
| QuRater-M | 0.6014 | 0.6912 | 0.7291 | 0.6927 | 0.7005 | 0.6703 | 0.6726 | 0.6633 | 0.5983 | 0.6471 | 0.6780 | 0.6912 | 0.6757 | 0.6269 | 0.5797 | 0.5875 | 0.6610 | 0.6881 |
| MuRater(M) | 0.6037 | 0.6989 | 0.7314 | 0.7074 | 0.7059 | 0.6989 | 0.6803 | 0.6649 | 0.6246 | 0.6656 | 0.6943 | 0.7098 | 0.6393 | 0.5820 | 0.6029 | 0.6811 | 0.6865 |
| MuRater(E) | **0.6231** | 0.7082 | 0.7307 | **0.7059** | 0.7012 | **0.6950** | 0.6834 | 0.6811 | 0.6416 | 0.6610 | 0.6927 | 0.6981 | **0.7144** | 0.6517 | 0.5967 | 0.6122 | 0.6850 | 0.6981 |

Table 14: Detailed per-language performance on across **BMLAMA**. Bold indicates the best result for each language.

| Method | AR | DE | EN | ES | FR | ID | IT | JA | KO | MS | NL | PT | RU | TH | TL | TR | VI | ZH |
|---|---|---|---|---|---|---|---|---|---|---|---|---|---|---|---|---|---|---|
| Uniform | 0.3128 | 0.4530 | 0.5148 | 0.4531 | 0.4563 | 0.3860 | 0.4422 | 0.4082 | 0.2764 | 0.3536 | 0.4001 | 0.4026 | 0.3391 | 0.2862 | 0.4702 | 0.3063 | 0.4294 | 0.4387 |
| QuRater-M | 0.3850 | 0.5540 | 0.5838 | 0.5075 | 0.4860 | 0.4757 | 0.5076 | 0.4317 | 0.3388 | 0.4629 | 0.5186 | 0.4835 | 0.3846 | 0.3431 | 0.5040 | 0.3971 | 0.4934 | 0.3931 |
| MuRater(M) | 0.4039 | 0.5660 | 0.6331 | 0.5725 | 0.5582 | 0.5544 | 0.5563 | 0.4353 | 0.3389 | 0.5364 | 0.5404 | 0.5194 | 0.4350 | 0.3679 | 0.5519 | 0.4393 | 0.5643 | 0.4500 |
| MuRater(E) | **0.4451** | **0.6062** | **0.6380** | **0.5919** | **0.5549** | **0.5898** | **0.5828** | **0.4749** | **0.3920** | **0.5703** | **0.5933** | **0.5578** | **0.4646** | **0.3828** | **0.5615** | **0.4576** | **0.6034** | **0.4669** |

Table 15: Detailed per-language performance on across **XCOPA**. Bold indicates the best result for each language.

| Method | ID | IT | TH | TR | VI | ZH |
|---|---|---|---|---|---|---|
| Uniform | 68.20 | 66.60 | 57.20 | 58.80 | **70.60** | 66.20 |
| QuRater-M | 65.20 | 65.00 | 56.00 | 58.40 | 67.40 | 65.80 |
| MuRater(M) | 67.80 | 67.20 | **58.20** | 58.80 | 69.00 | 65.60 |
| MuRater(E) | **69.00** | **68.20** | 57.20 | **60.20** | 70.20 | **69.60** |

Table 16: Detailed per-language performance on across **XNLI**. Bold indicates the best result for each language.

| Method | AR | DE | EN | ES | FR | RU | TH | TR | VI | ZH |
|---|---|---|---|---|---|---|---|---|---|---|
| Uniform | 35.90 | 46.47 | 47.67 | 45.74 | 46.14 | 43.29 | 38.35 | 39.60 | 39.56 | 40.60 |
| QuRater-M | **37.11** | 47.63 | 49.60 | **47.71** | 49.32 | 46.99 | 37.87 | 43.78 | **41.93** | 41.93 |
| MuRater(M) | 35.74 | 44.34 | 46.79 | 44.50 | 47.15 | 44.14 | **38.39** | 39.60 | 38.80 | 41.24 |
| MuRater(E) | 34.86 | **48.84** | **51.49** | 46.55 | **49.40** | **47.39** | 37.27 | **43.94** | 41.77 | **43.13** |

Table 17: Detailed per-language performance on **XWinograd**. Bold indicates the best result for each language.

| Method | EN | FR | JP | PT | RU | ZH |
|---|---|---|---|---|---|---|
| Uniform | **83.70** | **69.88** | **67.78** | 69.96 | 62.86 | **72.02** |
| QuRater-M | 77.12 | 66.27 | 66.21 | 66.54 | 60.32 | 63.49 |
| MuRater(M) | 78.54 | 65.06 | 67.47 | 67.30 | 62.86 | 67.86 |
| MuRater(E) | 80.22 | **69.88** | 66.32 | **71.48** | **65.71** | 68.25 |

(a) Translation from English (EN TO ML)

| Method | AR | DE | ES | FR | ID | IT | JA | KO | MS | NL | PT | RU | TH | TL | TR | VI | ZH |
|---|---|---|---|---|---|---|---|---|---|---|---|---|---|---|---|---|---|
| Uniform | 35.03 | 53.27 | 48.27 | 57.70 | 57.95 | 47.76 | 21.81 | 18.99 | 53.94 | 48.94 | 58.22 | 44.17 | 28.60 | 40.04 | 37.94 | 48.97 | 19.82 |
| QuRater-M | 36.60 | 52.70 | 48.28 | 58.74 | 59.62 | 48.17 | 23.59 | 19.52 | 54.23 | 48.20 | 59.03 | 45.09 | 29.94 | 42.05 | 39.73 | 48.65 | 19.97 |
| MuRater(M) | **37.84** | 53.65 | **48.92** | **59.45** | 60.17 | 48.85 | **24.01** | 21.26 | **54.33** | 49.51 | **60.30** | **46.70** | **30.78** | 41.83 | 40.11 | **50.89** | 19.98 |
| MuRater(E) | 37.80 | **53.87** | 48.30 | 58.85 | **60.20** | 49.39 | 23.73 | 20.99 | 54.03 | 49.52 | 60.05 | 46.14 | 29.84 | **42.49** | 40.64 | 50.79 | **20.40** |

(b) Translation to English (ML TO EN)

| Method | AR | DE | ES | FR | ID | IT | JA | KO | MS | NL | PT | RU | TH | TL | TR | VI | ZH |
|---|---|---|---|---|---|---|---|---|---|---|---|---|---|---|---|---|---|
| Uniform | 52.14 | **61.41** | **54.46** | 61.71 | 57.93 | 55.28 | **42.48** | 41.73 | 56.63 | **54.41** | 64.62 | 54.96 | 45.81 | **49.27** | 45.40 | 52.24 | **46.10** |
| QuRater-M | 51.14 | 60.40 | 53.63 | 61.47 | 57.81 | 55.68 | 42.13 | 41.04 | 55.82 | 53.77 | 64.34 | 53.30 | 44.44 | 47.39 | 46.58 | 51.23 | 44.62 |
| MuRater(M) | 52.39 | 60.86 | 54.26 | **62.64** | 57.77 | **56.21** | 42.47 | **42.47** | **56.71** | 54.26 | **64.80** | **55.00** | 45.65 | 48.84 | 46.17 | **52.53** | 45.40 |
| MuRater(E) | **52.63** | 60.93 | 54.03 | 61.72 | **57.98** | 56.00 | 42.25 | 41.48 | 56.12 | 54.23 | 64.59 | 54.55 | **46.01** | 47.97 | **47.03** | 52.04 | 45.31 |

Table 18: Detailed per-language performance on **FLORES**. Bold indicates the best result for each language.

| Method | AMMLU | CMMLU | INDOMMLU | JMMLU | VLMU |
|---|---|---|---|---|---|
| Uniform | 0.2594 | 0.3175 | 0.3235 | 0.3079 | 0.2909 |
| QuRater-M | 0.2659 | 0.3398 | 0.3278 | 0.3197 | 0.2898 |
| MuRater(M) | 0.2713 | **0.3467** | 0.3441 | 0.3304 | 0.3005 |
| MuRater(E) | **0.2714** | 0.3404 | **0.3489** | **0.3323** | **0.3048** |

Table 19: Detailed per-language performance on across **MMLU-L**. Bold indicates the best result per column.

### D.3 Impact of Translation and Data Quality on Multilingual Performance

To further investigate translation and data quality impact on multilingual performance, we analyze two representative language pairs with comparable token ratios but differing translation quality: *Japanese vs. Spanish* and *Thai vs. Korean*. Human evaluation results reveal that Japanese and Thai translations receive approximately 0.5 lower average quality scores than their respective counterparts, Spanish and Korean. As detailed in Tables above, this discrepancy is reflected in downstream performance, where Japanese and Thai consistently underperform across most multilingual benchmarks.

Table 20 presents the normalized top-10% selection scores (relative to Chinese). These results show that Japanese and Korean data exhibit notably lower selection scores than Spanish and Thai, aligning with observed translation-quality trends.

Table 20: Normalized top-10% document scores across languages (relative to Chinese).

| **Language** | ja | de | es | ar | id | pt | th | fr | vi |
|---|---|---|---|---|---|---|---|---|---|
| **Score** | 0.743 | 0.862 | 0.922 | 0.877 | 0.996 | 0.896 | 0.922 | 0.806 | 0.967 |
| **Language** | ms | tl | it | ko | ru | tr | zh | nl | |
| **Score** | 0.817 | 0.690 | 0.877 | 0.843 | 0.940 | 0.941 | 1.000 | 0.862 | |

These findings suggest that translation quality partially accounts for the observed performance gaps, yet it is not the sole determinant. Additional factors—including the intrinsic quality of the source corpora, language-specific tokenizer compression effects [15], and cross-lingual knowledge transfer dynamics [22]—likely contribute to the variation in multilingual model performance. Collectively, the results highlight that improving translation fidelity and ensuring balanced corpus quality are both critical for enhancing multilingual LLM training.

# E Case Study

We present examples from various languages exhibiting a range of quality scores. The results demonstrate that texts with higher scores tend to be more fluent and contain richer educational content, particularly in domains such as health and science. Moreover, for texts with comparable scores, the quality remains consistent across different languages. This suggests that our MuRater model evaluates text quality in a language-agnostic manner, relying solely on the content rather than the language in which it is written.

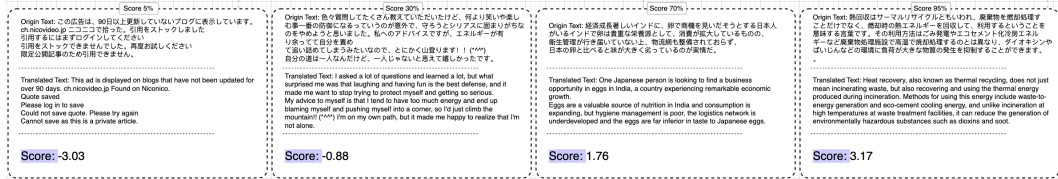

Figure 12: Sampled training examples of **Japanese** with quality ratings at different score range

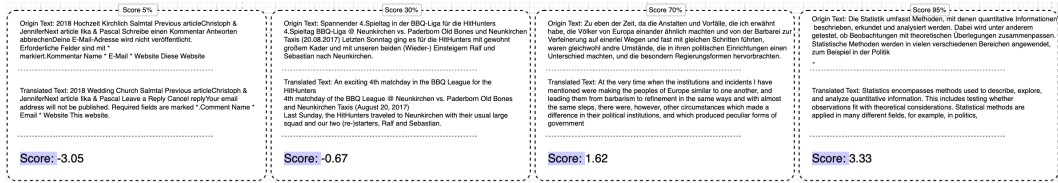

Figure 13: Sampled training examples of **German** with quality ratings at different score range

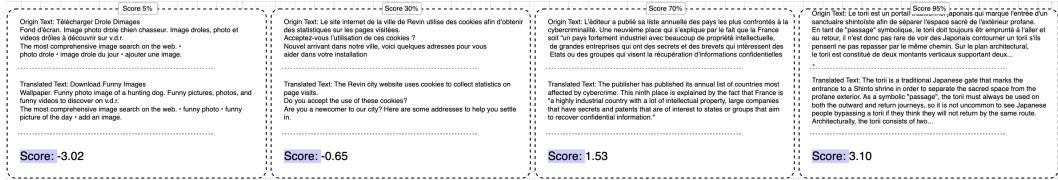

Figure 14: Sampled training examples of **French** with quality ratings at different score range

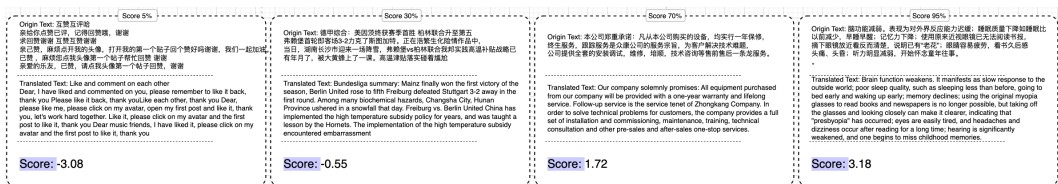

Figure 15: Sampled training examples of **Chinese** with quality ratings at different score range

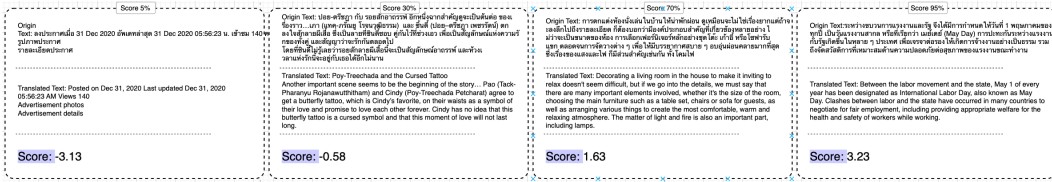

Figure 16: Sampled training examples of **Thai** with quality ratings at different score range

