# OpenReview forum: "MuRating: A High Quality Data Selecting Approach to Multilingual Large Language Model Pretraining"
_NeurIPS.cc/2025/Conference — NeurIPS 2025 poster_

### Official Review · Reviewer_erR2 · 2025-06-30

**Clarity:** 3
**Significance:** 3
**Originality:** 3
**Rating:** 4
**Confidence:** 3

**Summary:**

This work presents a novel method to curate high quality multilingual datasets based on transferring data-quality signals to other target languages. Multiple state-of-the-art English pairwise ratings are aggregated via a Bradley-Terry model into a single quality score. For the multilingual case the samples are translated to the respective languages and either re-rated or the rating is transferred. The data selection approach based on the aggregation of multiple quality scores shows promising performance in English-only as well as when transferred to the multilingual setup.

**Questions:**

- Could the authors please clarify the rationale behind the decision not to compare MuRatings performance with baselines such as HPLT-2, Fineweb-2 and Fineweb-HQ (for all the overlapping languages)?
- Could the authors provide more detailed information about the multilingual corpus used for LLM training? For example, it would be beneficial to know:
  - How was language identification performed?
  - What is the language mixture used (number of tokens by language) for the multilingual experiments?
  - How were the Fineweb filter adjusted for the multilingual context?
- Given that the classifier training dataset of FineWeb-HQ reportedly includes resources like the Aya collection (an instruct-style dataset available in 101 languages), could the authors clarify the statement in line [95] regarding the method's reliance on the availability of high-quality samples of multilingual benchmarks? There seems to be a potential discrepancy that warrants further explanation.

**Ethical Concerns:**

["NO or VERY MINOR ethics concerns only"]

**Final Justification:**

I thank the authors for the detailed rebuttal. They have addressed my concerns.

**Limitations:**

Yes

**Paper Formatting Concerns:**

No concern found.

**Quality:**

2

**Strengths And Weaknesses:**

**Strength:**
- The work presents a novel method to aggregate existing data quality raters in the English and multilingual context demonstrating promising performance. As high-quality multilingual data is lagging behind English data curation this novel method is an important contribution.
- The work and experiments are generally well structured and easy to follow.

**Weaknesses:**
- The work mentions Fineweb-HQ in the introduction, which has a significant overlap in languages. To critically discuss the method's performance in the context of existing multilingual work, I would have expected the work to compare to this baseline or at least to other multilingual baseline dataset, such as HPLT-2 and FineWeb-2.
- The work could be more detailed on the experimental setup, e.g. how was language identification performed or what is the language mixture used (number of tokens by language) for the multilingual experiments.
- The results of section 4.3.2. Translation Quality Assessment, could be strengthened with discussion including how translation performance impacts data selection and  thus LLM performance.

---

> ### Author Rebuttal · Authors · 2025-07-31
>
> We appreciate the reviewers' positive feedback on both the proposed methodology and paper writing. We have provided clarifications to comments and made several updates that have improved the paper based on your suggestions.
>
> ## Weaknesses and Questions
>
> > **Q1.** The work mentions Fineweb-HQ in the introduction, which has a significant overlap in languages. To critically discuss the method's performance in the context of existing multilingual work, I would have expected the work to compare to this baseline or at least to other multilingual baseline dataset, such as HPLT-2 and FineWeb-2.
>
> Thank you for the insightful suggestion. We have conducted additional experiments comparing our method with other multilingual baseline datasets, including FineWeb-HQ, FineWeb-2, and HPLT-2. Following the primary experimental setup in our paper, we train a 1.2B LLM on 200B English tokens and 300B multilingual tokens, keeping the language mix ratios consistent with our main experiment.
>
> Since FineWeb-HQ does not cover several languages in our setting, we limit the comparison to the 13 overlapping languages. For the remaining languages, we use the same data across the experiment to ensure a fair comparison.
>
> As shown in the results below, our method consistently outperforms all three baselines in multilingual data selection. Compared to FineWeb-2 and HPLT-2, our method achieves higher scores across all benchmarks and demonstrates stronger overall performance. Against FineWeb-HQ, our method outperforms it on nearly all benchmarks, with slightly lower results on FLORES and StoryCloze, which are narrative and translation tasks. We identify these as important future directions for improvement.
>
> These additional comparisons have been included in the paper to further validate the effectiveness of our approach in selecting high-quality multilingual data for LLM pretraining.
>
> |                  |MMLU_L|ARC_C|ARC_E|FLORES|HELLASWAG|MMLU_T|XCOPA|XNLI|StoryCloze|XWINOGRAD|BMLAMA|AVERAGE|
> |------------------|------|--------|--------|------|----------|------|-----|----|-----------|----------|------|--------|
> |MuRater|**31.96**|**35.01**|**58.98**|**47.27**|**47.11**|**29.40**|**65.73**|**44.46**|**67.67**|**74.13**|**52.97**|**50.43**|
> |HPLT-2|29.24|26.94|45.95|42.66|39.87|27.58|59.67|42.14|65.09|71.77|48.38|45.39|
> |FineWeb-2 |28.33|27.50|45.97|44.52|42.39|27.75|62.57|41.00|66.77|72.89|41.51|45.56|
> |----------------------|
> |MuRater(13lang) |**31.96**|**36.03**|**61.63**|50.11|**49.17**|**29.43**|**67.44**|**44.49**|68.75|**68.83**|**53.19**|**51.00**|
> |FineWeb-HQ(13lang)|30.52|30.91|54.37|**50.97**|46.15|28.66|64.92|41.79|**69.48**|68.72|43.08|48.14|
>
> > **Q2.** The work could be more detailed on the experimental setup, e.g. how was language identification performed or what is the language mixture used (number of tokens by language) for the multilingual experiments.
> (1)How was language identification performed?
> (2)What is the language mixture used (number of tokens by language) for the multilingual experiments?
> (3)How were the Fineweb filter adjusted for the multilingual context?
>
> Thank you for the helpful suggestion. We have added further details to clarify the experimental setup in the revised paper:
>
> **(1) Language Identification:**  We use the open-sourced FastText language identification model [1], which supports the detection of 176 languages and is widely adopted in large-scale multilingual data processing pipelines.
>
> **(2) Language Mixture:**  For multilingual data selection, we first select the top 10% highest-scoring documents for each language from our raw dataset. This results in a distribution closely aligned with the natural composition of Common Crawl and similar to that of FineWeb-2. The initial token distribution is:
> - ru (14.29%), es (12.83%), ja (12.19%), de (12.19%), zh (9.26%), fr (8.98%), it (7.29%), pt (4.58%), nl (4.53%), vi (3.21%), id (2.88%), ar (2.75%), tr (2.20%), th (1.51%), ko (1.41%), tl (0.04%), ms (0.02%)
>
> We then apply temperature-based sampling with $\tau=3.33$, following the common practice in multilingual LLM training [2]. The final token distribution used in our experiments is:
> - ru (8.08%), es (7.78%), ja (7.62%), de (7.62%), zh (7.00%), fr (6.93%), it (6.50%), pt (5.71%), nl (5.70%), vi (5.07%), id (4.86%), ar (4.78%), tr (4.52%), th (4.15%), ko (4.11%), tl (2.75%), ms (2.62%)
>
> **(3) FineWeb Filter Adjustments:**
> We adopt the filtering configuration of FineWeb-2, which applies heuristics such as thresholds on non alpha words ratio, minimum/maximum/average character lengths, and other quality signals. While the full configuration is available in the FineWeb-2 GitHub repository (not linked here due to NeurIPS 2025 anonymity policy), we adapt this setup to our own dataset for cost-efficiency reasons. This allows us to achieve comparable quality while keeping data processing and storage costs manageable at scale.
>
> > **Q3.** The results of section 4.3.2. Translation Quality Assessment, could be strengthened with discussion including how translation performance impacts data selection and thus LLM performance.
>
> Thank you for the suggestion. To explore how translation quality impacts data selection and downstream LLM performance, we conducted a supplementary experiment using the open-sourced LLM Qwen 3-8B, which has a FLORES chrF score of 56 and lower than GPT-4o’s score, 62. We used both models to translate our preference training pairs and trained separate MuRater models.
>
> Each MuRater was used to score 10,000 multilingual documents per language. We then compared their annotation results using Pearson correlation and Kendall’s Tau. As shown below, the two models produce highly consistent annotations across all languages. This suggests that even with moderately degraded translation quality, MuRater can still be effectively trained—as long as the relative preference score of the training pairs is preserved—highlighting the robustness of our approach.
>
> |Language|ja|de|es|ar|id|pt|th|fr|vi|it|ko|ms|tl|ru|tr|zh|nl|
> |--------|--|--|--|--|--|--|--|--|--|--|--|--|--|--|--|--|--|
> |**Kendall’s Tau**|0.8930|0.9005|0.8951|0.8645|0.8903|0.8964|0.8834|0.8924|0.8867|0.8851|0.8885|0.8427|0.8749|0.8942|0.8822|0.9044|0.8991|
> |**Pearson Correlation**|0.9835|0.9854|0.9843|0.9793|0.9822|0.9848|0.9803|0.9831|0.9825|0.9815|0.9821|0.9644|0.9755|0.9850|0.9818|0.9861|0.9843|
>
>
> To further analyze language-level variation, we focus on two language pairs with similar token ratios but differing translation quality: Japanese vs Spanish and Thai vs Korean. Human evaluations show that Japanese and Thai have ~0.5 lower translation scores than Spanish and Korean. As shown in Tables 6–13 (Appendix D.2), this is reflected in LLM performance—Japanese and Thai consistently underperform their counterparts across most benchmarks.
>
> We also present the normalized score of selected data assigned by MuRater (relative to zh):
> |Language| ja| de | es | ar | id| pt | th|fr|vi|ms|tl|it|ko|ru|tr|zh|nl|
> |----------|------|------|------|------|------|------|------|------|------|------|------|------|------|------|------|------|------|
> |Score|0.743|0.862|0.922|0.877|0.996|0.896|0.922|0.806|0.967|0.817|0.690|0.877|0.843|0.940|0.941|1.000|0.862|
>
> Interestingly, these results indicate that Japanese data tend to be of lower quality than Spanish, but Thai data have higher quality than Korean, which is supported by the uniform selection results in Tables 6–13 (Appendix D.2). The reverse performance between Thai and Korean suggest that while translation quality may influence data selection and LLM performance, other factors-such as corpus original quality, tokenizer compression ratio of different languages [3], and cross-lingual knowledge transfer [4]—also play significant roles.
>
> We view this as an important direction for future work. We plan to further explore the reason of language performance imbalance from translation quality, tokenizer compression ratio and cross-lingual knowledge transfer. We will also continue to invest in human translation and evaluation to enhance the quality of multilingual data selection.
>
> > **Q4.** Given that the classifier training dataset of FineWeb-HQ reportedly includes resources like the Aya collection (an instruct-style dataset available in 101 languages), could the authors clarify the statement in line [95] regarding the method's reliance on the availability of high-quality samples of multilingual benchmarks? There seems to be a potential discrepancy that warrants further explanation.
>
> Thank you for pointing this out. Our original statement was based on the observation that some multilingual benchmarks like MMMLU [5] cover only a limited set of languages (e.g., 14), leading us to consider that certain high-quality benchmarks were unavailable for many others. Upon further consideration, we agree that this limitation can be mitigated through high-quality translation. To avoid confusion, we have removed the statement from the paper.
>
> We also recognize FineWeb-HQ as a valuable contribution to multilingual pretraining data selection. As demonstrated in our comparative experiments above, it shows strengths in narrative task such as StoryCloze and translate  task Flores. We appreciate its contributions and plan to explore how the strengths of both approaches can be combined in future work to advance multilingual data selection and support more effective LLM pretraining across languages.
>
> [1] Joulin, A., et al. "FastText.zip: Compressing Text Classification Models." arXiv:1612.03651, 2016.
> [2] Xue, L., et al. "mT5: A Massively Multilingual Pre-trained Text-to-Text Transformer." NAACL 2021.
> [3] Goldman, O., et al. "Unpacking Tokenization: Evaluating Text Compression and its Correlation with Model Performance." ACL Findings, 2024.
> [4] He, Y., et al. "Scaling Laws for Multilingual Language Models." arXiv:2410.12883, 2024.
> [5] OpenAI. MMMLU: Multilingual Massive Multitask Language Understanding [Dataset]. Hugging Face. Accessed 31 July 2025.

---

> > ### Comment · Reviewer_erR2 · 2025-08-03
> >
> > I thank the authors for the detailed rebuttal and addressing my concerns.

---

> > > ### Author Response · Authors · 2025-08-04
> > >
> > > Dear Reviewer erR2,
> > >
> > > Thank you once again for your thoughtful and constructive feedback on our work.
> > >
> > > If there are any additional questions or clarifications you would like us to address during the rebuttal phase, we would be more than happy to respond.
> > >
> > > Your insights have been truly helpful and have contributed to improving the quality of our paper.
> > >
> > > Best regards,
> > >
> > > All authors

---

### Official Review · Reviewer_Vx9U · 2025-07-03

**Clarity:** 3
**Significance:** 3
**Originality:** 3
**Rating:** 4
**Confidence:** 3

**Summary:**

This paper introduces MuRating, a scalable data selection framework that leverages high-quality English data signals to build a multilingual rater across 17 languages. The proposed method consist of multiple steps such as 1) Aggregating multiple English "raters" via pairwise comparisons to learn a unified document-quality score, 2) Transferring this judgement through translation data pairs to train a multilingual quality evaluator, then 3) applying the evaluator to web-crawled mutlingual texts to select high-quality data.
In the experiment, they pretrained a 1.2B parameter LLaMA model using the selected multilingual corpus. Results show substantial improvements over strong baselines like QuRater, Ask LLM and DCLM, particularly on knowledge-intensive tasks.

- this paper is well motivated and mostly easy to follow.
- Tables are too small to read. Please consider making it bigger in the camera-ready version.

**Questions:**

see the summary.

**Ethical Concerns:**

["NO or VERY MINOR ethics concerns only"]

**Final Justification:**

In response from the author, my questions/concerns are addressed. I increased my score to 4.

**Quality:**

3

**Strengths And Weaknesses:**

strengths
- the authors exploits English-quality signals and translation to bootstrap evaluators for low-resourced languages. This help offer a practical path for multilingual data curation.

weaknesses
- there could be selection bias. The proposed approach relies on English-originated quality criteria, so this may not capture culturally or linguistically relevant features in non-English language texts. Have you ever conducted some analysis in this line?
- Experiments are limited to a 1.2B model. It might be still uncertain how the proposed approach scale to larger architectures or more diverse data settings. If you could run additional experiments with e.g., 7B model if the authors have enough computational resources, that'd be great.

---

> ### Author Rebuttal · Authors · 2025-07-31
>
> Thanks for your positive comments on our method and paper-writing. We have provided clarifications to your comments and made several updates that can improve the paper based on your suggestions.
> ## Weakness
> > **Q1.** there could be selection bias. The proposed approach relies on English-originated quality criteria, so this may not capture culturally or linguistically relevant features in non-English language texts. Have you ever conducted some analysis in this line?
>
> Thank you for highlighting this important concern. We would like to clarify that the motivation behind our method is to transfer general quality criteria—such as educational value, coherence, and fluency, which have been shown to be effective in English data selection [1,2], but are not inherently English-specific. Given the limited work on multilingual data selection, our goal is to establish a strong, general-purpose foundation as a first step toward broader multilingual data filtering.
>
> To further address potential cultural or linguistic bias, we included a baseline called QuRater-M (Lines 205–206), which leverages a strong multilingual model, GPT-4o, to directly score preference pairs in their original languages. We consider this approach can capture culturally and linguistically specific features inherent to each language by leveraging the multilingual ability of GPT-4o. As shown in Table 1, our method outperforms QuRater-M on tasks such as MMLU-L, which includes culturally specific local questions, and FLORES, which tests linguistic knowledge through translation. In our additional 7B experiments (shown below), we further evaluate on the MultiLoKo [3] benchmark, which includes tasks that are locally relevant to specific languages as well as translation-based tasks. MuRater continues to perform strongly, suggesting that the general quality criteria we apply are also effective in capturing culturally and linguistically relevant content.
>
> We fully agree that language-specific selection is important and consider it a key direction for future work, following our establishment of a solid general-purpose data selection criterion for multilingual data. We plan to explore language-specific criteria to better identify culturally and linguistically tailored content, aiming to advance multilingual language model pretraining in the future.
>
> > **Q2.** Experiments are limited to a 1.2B model. It might be still uncertain how the proposed approach scale to larger architectures or more diverse data settings. If you could run additional experiments with e.g., 7B model if the authors have enough computational resources, that'd be great.
>
> Thank you for the valuable suggestion. To evaluate the robustness of our approach, we conducted additional experiments using a 7B model with an architecture similar to the 1.2B model discussed in the main paper. In this setting, we compared the performance of MuRater and baseline QuRater-M.
>
> The model was trained on 1T tokens, with 16.5% multilingual data selected by each method. The remaining 83.5%, consisting of English, code, and math data, was kept identical across both setups. The language mix ratio for the multilingual portion follows the same distribution as in our main experiments.
>
> We chose this setup—with a larger model architecture, a lower multilingual data ratio, and a more diverse mix of data types—to evaluate the robustness of our method under varied training conditions. As shown in the results below, MuRater consistently outperforms QuRater-M across most tasks, demonstrating its effectiveness in multilingual data selection when scaled to larger models and more diverse data.
>
> Notably, the strong performance on MMLU_L, FLORES, and MultiLoKo further validates that our general quality criteria are capable of capturing culturally and linguistically relevant content, thereby enhancing multilingual LLM performance.
>
> We have included these results in our paper to strengthen the empirical support for MuRater. Additionally, we plan to open-source MuRater to support the research of multilingual data selection and encourage further evaluation and improvement by the broader research community.
>
> |              | MMLU_L| ARC_C| ARC_E| FLORES| HELLASWAG| MMLU_T| XCOPA | XNLI | STORYCLOZE | XWINOGRAD |BMLAMA| MultiLoKo |Average|
> |------------|-------|----------|----------|--------|------------|--------|--------|------|------------|----------|--------|---------|---------|
> | QuRater-M|35.93| 42.36  | 65.20  | 55.15  | 57.29      | 32.54  | 69.27  | **45.13** | 74.50   | 82.60 | 49.61  | 8.87       | 51.54   |
> | MuRater| **36.87**| **43.38**| **66.87**| **55.38**| **57.76**    | **32.93**| **71.03**| 45.06 | **75.42**      | **83.19**    | **52.82**| **10.61**  | **52.61** |
>
> > **Suggestion** Tables are too small to read. Please consider making it bigger in the camera-ready version.
>
> Thank you for the helpful suggestion. We will adjust the table formatting and enlarge the content to improve readability in the camera-ready version.
>
> [1] Wettig, Alexander, et al. "QuRating: Selecting High-Quality Data for Training Language Models." International Conference on Machine Learning. PMLR, 2024.
> [2] Li, Jeffrey, et al. "Datacomp-lm: In search of the next generation of training sets for language models." Advances in Neural Information Processing Systems 37 (2024): 14200-14282.
> [3] Hupkes, Dieuwke, and Nikolay Bogoychev. "MultiLoKo: a multilingual local knowledge benchmark for LLMs spanning 31 languages." arXiv preprint arXiv:2504.10356 (2025).

---

> ### Author Response · Authors · 2025-08-05
>
> Dear Reviewer Vx9U,
>
> Thank you once again for your valuable and constructive feedback on our work.
>
> We truly appreciate the time and effort you have invested in reviewing our paper. If there are any further aspects you'd like us to clarify or elaborate on during the discussion period, we would be glad to provide additional details.
>
> Your suggestions have been instrumental in helping us refine our work. We hope the improvements we’ve made have addressed your concerns and are reflected in your overall assessment.
>
> Best regards, All authors

---

### Official Review · Reviewer_xazu · 2025-07-03

**Clarity:** 4
**Significance:** 3
**Originality:** 3
**Rating:** 5
**Confidence:** 3

**Summary:**

In this paper, the authors proposed Murating, a scalable framework for multilingual LLM pretraining that transfers high-quality English data-selection signals to 17 target languages. Existing model-based data selection methods are primarily English-centric, lacking robust multilingual adaptation. MuRating uses unifying English Raters, Translation-Based Multilingual transfer and Cross-lingual regularization to achieve robust multilingual adaptation.

**Questions:**

Please see Weaknesses.

**Ethical Concerns:**

["NO or VERY MINOR ethics concerns only"]

**Final Justification:**

All of my concers have been addressed, I hence decide to raise my score.

**Limitations:**

Many LLMs are included as raters in the proposed method, so the fairness of the comparisons is a concern.

**Quality:**

3

**Strengths And Weaknesses:**

Strengths:

(1) The paper is well-written and easy to follow;

(2) MuRating extends data selection to low-resource languages without requiring language-specific annotations;

(3) Aggregating various English rates via pairwise comparisons mitigates individual rater biases.

(4) Parallel text pairs enforce semantic alignment, reducing score divergence.

(5) The experimental results are promising.

Weaknesses:

(1) The translation quality is heavily reliant on LLM translators.

(2) How do the authors guarantee the translation quality, Did the authors use human verifiers? The authors should provide details and discussions on this aspect, as LLM translators are not very skilled in low-resource languages.

(3) As MuRating utilizes multiple LLM as raters, are the comparisons unfair, the  authors should provide discussions on the fairness of the experimental comparison shown in this paper.

---

> ### Author Rebuttal · Authors · 2025-07-31
>
> We appreciate the reviewers' positive feedback on the methodology and writing of our paper. In response to the provided comments, we have made clarifications and implemented several new improvements that further strengthen our work.
>
> ## Weakness
> > **Q1.** The translation quality is heavily reliant on LLM translators
>
> Thank you for the observation. While our method relies on LLMs for translation, we conducted a human evaluation to ensure translation quality, as detailed in Lines 292–304. For each language, we randomly sampled 50 source-translation pairs and had them rated on a 1–5 scale by human experts with CEFR C1-level proficiency or above in both the target language and English. A score of 3 indicates acceptable translation quality. The evaluation cost approximately $1200 (Appendix C.2, page 19). As shown in Figure 5, most languages received average scores above 4, while Japanese and Thai scored above 3.5, indicating consistently good translation quality.
>
> We acknowledge the importance of maintaining high-quality translations and will continue to invest in human translation and evaluation to further improve translation quality. Additionally, we plan to open-source our annotated training dataset to encourage community participation and advance multilingual data selection for LLM pretraining.
>
>
> > **Q2.** How do the authors guarantee the translation quality, Did the authors use human verifiers?
>
> Thank you for raising these important questions. As detailed in Lines 292–304, we conducted a human evaluation to assess translation quality. For each language, we randomly sampled 50 source-translation pairs and asked human experts to rate them on a 1–5 scale, where a score of 3 indicates acceptable quality. All evaluators held CEFR C1-level proficiency or above in both the target language and English. The evaluation cost approximately $1200, as noted in Appendix C.2 (page 19). As shown in Figure 5, most languages achieved average scores above 4, while Japanese and Thai scored above 3.5, indicating consistently good translation quality across all languages.
>
> We are committed to further improving translation quality by increasing human expert involvement and plan to open-source our annotated training dataset. By engaging the research community in dataset construction, we aim to advance multilingual data selection practices for LLM pretraining.
>
> In addition, we conducted a supplementary experiment using the open-sourced LLM Qwen 3-8B, which has a FLORES chrF score of 56—lower than GPT-4o’s score of 62 for the same language. We used both Qwen 3-8B and GPT-4o to translate our training preference pairs and trained separate MuRater models accordingly. Each MuRater was then used to score 10,000 multilingual documents per language. We compared the annotated scores between the two models using Pearson correlation and Kendall’s Tau. As shown below, both metrics indicate high agreement between the two annotation sets across all languages.
>
> | Language | ja     | de     | es     | ar     | id     | pt     | th     | fr     | vi     | it     | ko     | ms     | tl     | ru     | tr     | zh     | nl     |
> |----------|--------|--------|--------|--------|--------|--------|--------|--------|--------|--------|--------|--------|--------|--------|--------|--------|--------|
> | **Kendall’s Tau**        | 0.8930 | 0.9005 | 0.8951 | 0.8645 | 0.8903 | 0.8964 | 0.8834 | 0.8924 | 0.8867 | 0.8851 | 0.8885 | 0.8427 | 0.8749 | 0.8942 | 0.8822 | 0.9044 | 0.8991 |
> | **Pearson Correlation**  | 0.9835 | 0.9854 | 0.9843 | 0.9793 | 0.9822 | 0.9848 | 0.9803 | 0.9831 | 0.9825 | 0.9815 | 0.9821 | 0.9644 | 0.9755 | 0.9850 | 0.9818 | 0.9861 | 0.9843 |
>
> These results suggest that even with a weaker translation model like Qwen 3-8B, as long as the relative preference between the training pairs are preserved, MuRater can still be effectively trained. This highlights the robustness of our approach to moderate variations in translation quality.
>
>
> > **Q3.** As MuRating utilizes multiple LLM as raters, are the comparisons unfair, the authors should provide discussions on the fairness of the experimental comparison shown in this paper.
>
> Thank you for raising this important concern. We consider there may be a misunderstanding, and we would like to clarify that MuRating does **not** use multiple LLMs as raters during inference or data selection. As described in Lines 110–114 of Section 3.1, we use multiple LLMs **only during training data construction**, where they are asked to score and compare preference pairs. This results in a training dataset used to fine-tune a **single encoder-based LLM with a linear head**, referred to as **MuRater** (Lines 186–189).
>
> In all experiments, the **single MuRater**—like other baseline raters—is used to score the same corpus and select the top 10% of documents. The selected data is then used to train a 1.2B LLM from scratch. This ensures a consistent evaluation setup across methods and a fair comparison. As shown in Tables 1 and 2, MuRater performs effectively for both multilingual and English data selection under these controlled and fair comparison settings.
>
> We appreciate the reviewer’s comment and revised the experimental setup section to clearly emphasize that only one rater is used during selection to avoid further confusion.

---

> > ### Comment · Reviewer_xazu · 2025-08-06
> >
> > I thank the authors for their detailed responses. I appreciate their efforts to add more experimental evidence to support their claim. After carefully checking their responses, I think all of my concerns have been addressed, I hence decide to raise my score.

---

> > > ### Author Response · Authors · 2025-08-08
> > >
> > > Thank you very much for taking the time to review our rebuttal. We are glad to hear that it has addressed your main concerns, and we truly appreciate your valuable suggestions, which have helped us significantly improve the manuscript.

---

> ### Author Response · Authors · 2025-08-05
>
> Dear Reviewer xazu,
>
> Thank you once again for your valuable and constructive feedback on our work.
>
> We truly appreciate the time and effort you have invested in reviewing our paper. If there are any further aspects you'd like us to clarify or elaborate on during the discussion period, we would be glad to provide additional details.
>
> Your suggestions have been instrumental in helping us refine our work. We hope the improvements we’ve made have addressed your concerns and are reflected in your overall assessment.
>
> Best regards, All authors

---

### Note · Authors · 2025-08-13

Dear Chairs and Reviewers,

We sincerely thank you for your time, effort, and dedication in overseeing and contributing to the review process. We truly appreciate your support in evaluating our responses and the improvements made to our paper. Your efforts mean a great deal to us. For everyone’s convenience, we would like to provide a summary of the reviews and our rebuttal.

The reviews recognized the significance of our work in extending English-only data raters to the multilingual setting for LLM pretraining data selection. Our proposed approach advances the field by effectively improving the performance of pretrained LLMs on multilingual tasks, particularly those requiring extensive knowledge. The paper was also commended for its clear structure, logical flow, well-designed experiments, and effective data selection methodology.

To address the main concerns raised, we made the following additions in our rebuttal:

- **Additional baselines** (Reviewer erR2)
  We expanded our experiments by including comparisons with other strong multilingual baselines: FineWeb-HQ, FineWeb-2, and HPLT-2. The results consistently show that our proposed MuRater outperforms these baselines, further validating its effectiveness.

- **Effect of translation quality** (Reviewers xazu, Vx9U, and erR2)
  We examined the influence of translation quality by re-running the data selection process using a weaker open-source model Qwen-3-8B for translation. The results demonstrated a high level of consistency with those obtained using our GPT-4o-based approach, indicating the robustness of our method against variations in translation quality.

- **Detailed information about the multilingual corpus** (Reviewer erR2)
  We provided additional clarifications and descriptions of the multilingual corpus used in our experiments to make the experimental setup clearer and more transparent in the revised manuscript.

- **Scalability and robustness**  (Reviewer Vx9U)

  We conducted further validation using a 7B model across a wider range of datasets. The findings confirm that our approach scales effectively to larger model sizes and is applicable across diverse data scenarios.

We have incorporated these changes into our revised paper to further strengthen our work. We sincerely appreciate the constructive feedback from all reviewers and look forward to future opportunities for continued discussion and collaboration.

Best regards,

All authors

---

### Decision · Program_Chairs · 2025-09-17

**Decision:**

Accept (poster)

**Comment:**

The paper proposes MuRating, a multilingual data selection framework that transfers quality signals from English raters into a unified multilingual evaluator across multiple languages. The method enables effective data curation for pretraining large language models, as validated by the empirical results. Reviewers highlighted the clarity of writing, the practical significance of addressing multilingual data curation, and the robustness of results to translation quality and model scale. Weaknesses include potential bias from English-originated quality signals and the limited coverage of cultural/linguistic features. During rebuttal, the authors provided new baselines, translation robustness studies, and new experiments, which satisfied most reviewer concerns.

Overall, this paper makes a meaningful and well-substantiated contribution to multilingual LLM pretraining. I recommend acceptance.